

# Nonadjacent dependencies and sequential structure of chimpanzee action during a natural tool-use task

Elliot Howard-Spink[1,2], Misato Hayashi[3,4], Tetsuro Matsuzawa[3,5,6], Daniel Schofield[7,8], Thibaud Gruber[9] and Dora Biro[10]

[1] Department of Biology, University of Oxford, Oxford, United Kingdom
[2] Development and Evolution of Cognition Group, Max Planck Institute of Animal Behavior, Konstanz, Germany
[3] Chubu Gakuin University, Kakamigahara, Gifu, Japan
[4] Japan Monkey Centre, Inuyama, Japan
[5] Division of the Humanities and Social Sciences, California Institute of Technology, Pasadena, California, United States
[6] College of Life Sciences, Northwest University, Yangling, Shaanxi, China
[7] School of Anthropology and Museum Ethnography, University of Oxford, Oxford, United Kingdom
[8] Visual Geometry Group, Department of Engineering, University of Oxford, Oxford, United Kingdom
[9] Faculty of Psychology and Educational Sciences, and Swiss Center for Affective Sciences, University of Geneva, Geneva, Switzerland
[10] Department of Brain and Cognitive Sciences, University of Rochester, Rochester, United States

Corresponding authors
Elliot Howard-Spink,
elliot.howardspink@outlook.com
Thibaud Gruber,
Thibaud.Gruber@unige.ch

## ABSTRACT

Many of the complex behaviours of humans involve the production of nonadjacent dependencies between sequence elements, which in part can be generated through the hierarchical organization of sequences. To understand how these structural properties of human behaviours evolved, we can gain valuable insight from studying the sequential behaviours of nonhuman animals. Among the behaviours of nonhuman apes, tool use has been hypothesised to be a domain of behaviour which likely involves hierarchical organization, and may therefore possess nonadjacent dependencies between sequential actions. However thus far, evidence supporting hierarchical organization of great-ape tool use comes from methodologies which have been criticised in their objectivity. Additionally, the extent to which nonadjacent dependencies appear in primate action sequences during tool use has not been formally investigated. We used estimations of mutual information (MI)—a measure of dependency strength between sequence elements—to evaluate both the extent to which wild chimpanzees produce nonadjacent dependencies during a naturalistic tool-use task (nut cracking), as well as how sequences of actions are likely organized during tool use. Half of adult chimpanzees produced nonadjacent dependencies at significantly greater sequential distances than comparable, nonhierarchical Markov models once repeated actions had been accounted for. Additionally, for the majority of chimpanzees, MI decay with increasing sequential distance included a power-law relationship, which is a key indicator that the action sequences produced by chimpanzees likely entail some degree of hierarchical organization. Our analysis offered the greatest support for a system of organization where short subroutines of actions (2–8 actions long) are hierarchically arranged into longer sequences—a finding which is consistent with previous qualitative descriptions of ape tool-use

behaviours. Interindividual variability was detected within our analysis in both the maximum distance dependencies were detected, and the most likely structuring mechanism for sequential action organization. We discuss these results in light of possible interindividual variation in the systems of action organization used by chimpanzees during tool use, in addition to methodological considerations for applications of MI estimations to sequential behaviours. Moreover, we discuss our main findings alongside hypotheses for the coevolution of complex syntax in language and tool-action across hominin evolutionary history.

# INTRODUCTION

Many behaviours of both humans and non-human animals (henceforth animals) can be considered sequences of discrete actions. Therefore, understanding how both humans and animals generate, store and retrieve such sequences is key to understanding the cognition supporting these serial behaviours. Humans in particular exhibit a number of sequential behaviours which are argued to be uniquely complex in comparison to the analogous behaviours of animals, including language (*Hauser, Chomsky & Fitch, 2002*; *Bolhuis et al., 2014*; *Hauser et al., 2014*; *Berwick & Chomsky, 2015*); the production and appreciation of music (*Fitch, 2013*; *Fitch & Martins, 2014*); and the manufacture and use of highly sophisticated tools (*Greenfield, 1991*; *McGrew, 1993*; *Davidson & McGrew, 2005*; *Vaesen, 2012*). Comparative research into the sequential behaviours of animals may offer key insights into the evolution of the elaborate structural features of human behaviours, particularly through the investigation of behaviours which are analogous to human communication, music and tool use (*Matsuzawa, 1996*; *Hauser, Chomsky & Fitch, 2002*; *Fitch & Hauser, 2004*; *Berwick et al., 2012*; *Hayashi, 2015*; *Sainburg et al., 2019*).

A key feature of human sequential behaviours is the emergence of nonadjacent dependencies. Nonadjacent dependencies emerge in sequences when elements co-occur with each other, despite being separated by intervening elements of a sequence (*Lin & Tegmark, 2017*; *Sainburg et al., 2019*; *Watson et al., 2020*). For example, in language, nonadjacent dependencies can emerge through subject-verb agreements, where the nature of the subject (gender, number, *etc.*) may influence the morphology of its corresponding verb, irrespective of how far separated they are in linear space and/or time (*Mel'čuk, 1988*; *Rispens & Soto de Amesti, 2017*). Of the various ways nonadjacent dependencies emerge in human behaviours, one commonly cited mechanism is through the hierarchical organization of sequences. During hierarchical organization, the atomic units of a behaviour (*e.g.*, phonemes, musical notes, and manual actions) are organized into short subroutines (*e.g.*, words, musical phrases, short goal-directed action sequences) which are themselves organized to produce wider strings of behaviour. Importantly, hierarchical organization can produce nonadjacent dependencies through embedding subroutines within each other. For example, in language, a clause can be embedded within another

clause (*Lashley, 1951*; *Chomsky, 1956*, *2002*; *Jäger & Rogers, 2012*; *Fitch & Martins, 2014*; *Sainburg et al., 2019*; *Sainburg, Mai & Gentner, 2022*); *e.g.*, in the sentence "*The weather is horrible!*", a prepositional phrase may be embedded to produce the sentence "*The weather [outside the window] is horrible!*". In this example, the verb 'is' and adjective 'horrible' both relate to the noun at the head of the phrase (the weather), despite being closer to another noun (the window) in linear space and/or time. Likewise during sequential actions, nonadjacent dependencies occur when a series of actions addressing one subgoal are embedded within a sequence of actions addressing another goal. For example, when opening a door an actor may grasp, turn, and push the handle. However, if the door is locked, the actor may grasp the handle, then begin eliciting a number of actions to open the lock (thus completing the embedded subgoal: to undo the lock), before continuing to turn and push the handle to open the door. In such an example, grasping a handle always precedes turning a handle, and therefore these actions are linked within the sequence; however, they may be separated by an intervening embedded string of actions. Whilst debate continues surrounding the extent to which language and action share homologous systems of organization (see *Coopmans, Kaushik & Martin, 2023* for a recent argument for the nuanced differences between these behaviours), the general ability to comprehend hierarchically organized information, and account for nonadjacent dependencies, are undoubtedly important for the performance of many complex human behaviours (*Lashley, 1951*; *Chomsky, 1956*, *2002*; *Rosenbaum et al., 2007*; *Jäger & Rogers, 2012*; *Fitch & Martins, 2014*).

Whilst nonadjacent dependencies in human behaviours—including those produced by hierarchical organization—have been subject to many decades of study, comparatively less research has assessed whether animals exhibit similar structural properties in the sequential behaviours that they habitually perform in the wild. Until relatively recently, animal sequential behaviours were assumed to be produced through reflexive serial chains: when an action or call was externalised, it would trigger the externalization of the next action or call, and so on to produce an extended behavioural sequence (*Hauser, Chomsky & Fitch, 2002*; *Terrace, 2005*; *Ten Cate & Okanoya, 2012*). The rules which animals employed to create such behavioural sequences were therefore theorized to be extremely simple, requiring no knowledge of the calls or actions they produced other than the one produced most recently (thus, significantly reducing the burden on working memory due to the absence of long-range nonadjacent dependencies; *Wilson et al., 2020*). Such reflexive chains can be characterized using simple stochastic modelling approaches, such as Markov chains: network models where each node represents a different possible element in a sequence, and the edges between nodes are weighted based on the likelihood a sequence will contain an adjacent transition between element types (*Gagniuc, 2017*). Despite the historic pervasiveness of chaining models for explaining the sequential structure of animal behaviours, mounting evidence suggests that a number of different animal species produce behavioural sequences which, to varying degrees, contain hierarchical patterns. For example, hierarchical organization is found the songs of whales (*Suzuki, Buck & Tyack, 2006*; *Miksis-Olds et al., 2008*) and birds (*Sainburg et al., 2019*), as well as in the behaviours of several primate species, including perceptions of grouped, hierarchical social

relationships in baboons (*Bergman et al., 2003*); the nested temporal organization of vocal elements during orangutan long calls (*Lameira et al., 2024*); and the mental organization of social games in chimpanzees (*Mielke & Carvalho, 2022*). For behaviours which are temporally organized, such as in the case of birdsong, these hierarchical patterns have been supported through the identification of nonadjacent dependencies, which span across sequential distances that match those predicted by hierarchical organization (*Sainburg et al., 2019*).

Nevertheless, nonadjacent dependencies and hierarchical organization do not appear to be universal aspects of animal behaviours. For instance, the communication systems of many species can be readily approximated using Markovian, non-hierarchical dynamics (*Hauser, Chomsky & Fitch, 2002*; *Fitch, Hauser & Chomsky, 2005*; *Ten Cate & Okanoya, 2012*), and do not result in nonadjacent dependencies over long distances, suggesting that the use of supraregular grammars to construct communicative sequences may be limited to animal species that employ complex broadcast signals. Additionally, some behaviours can employ different systems of organization at differing temporal scales. This is even the case for humans. For example in language, whilst words are hierarchically organized based on the rules of a language's syntax, the relationships between sound units within words (phonemes) can be characterized using their neighbouring associations (*Kaplan & Kay, 1994*; however see instances of nonadjacent dependencies through vowel harmony, such as *Gonzalez-Gomez et al., 2019*). Therefore, further research is required to understand whether animals are able to produce sequences which share similar structural features to those of humans, and if so, to identify the behavioural contexts in which these sequences are employed. This includes research into the behaviours of species which are phylogenetically proximate to humans—such as chimpanzees, bonobos, and other great apes—for whom these systems of sequential organization, if present, would have the highest likelihood of sharing homologous origins with similar behaviours of humans (*Gruber & Clay, 2016*).

Among the behaviours of wild chimpanzees—one of the most closely related great ape species to humans—tool use has been identified as a domain of behaviour which likely possesses high hierarchical complexity (*Byrne & Russon, 1998*; *Byrne, Sanz & Morgan, 2013*; *Gontier, 2024*), and consequentially, likely entails some form of nonadjacent-dependency generation. Previous descriptions of chimpanzee tool-use behaviours have included the possibility that some particularly complex forms of tool use require chimpanzees to understand hierarchical relationships between objects (*Matsuzawa, 1991*, *1996*; *Hayashi, 2007*; *Sanz & Morgan, 2010*; *Hayashi & Takeshita, 2022*). Moreover, when considering how chimpanzees organize their behavioural sequences for tool use, previous models used qualitative descriptions of the stages of producing, using, and disposing of tools, over many hours of human observation (the *chaîne opératoire*; *Byrne & Russon, 1998*; *Tonooka, 2001*; *Carvalho et al., 2008*; *Sanz & Morgan, 2010*; *Estienne, Stephens & Boesch, 2017*; *Boesch et al., 2020*). These observations have been used to suggest that, when engaging in tool-use behaviours, chimpanzees (and other apes) generate behavioural sequences hierarchically, through the decomposition of tool-use behaviours into a series of nested subgoals, which are themselves addressed

through the generation of short subroutines of manual actions (*Byrne & Russon, 1998*; *Byrne, Sanz & Morgan, 2013*). Hierarchical organization is theorized to offer an adaptive advantage for successful tool-use behaviours in comparison to associative chaining, as organizing sequences hierarchically would allow for apes to flexibly reorganize subgoals of behaviours based on the local context of their environments, rather than repeatedly following a 'fixed action pattern' (*Byrne, Sanz & Morgan, 2013*). For example when organizing actions hierarchically, an individual may omit optional steps (a chimpanzee may not need to puncture a hole in a termite mound if one already exists; *Byrne, Sanz & Morgan, 2013*), or through embedding extra steps within a behavioural sequence (*e.g.*, an individual may interrupt a behavioural sequence to adjust or swap over tool items, which are not necessary for the behavioural sequence, but may help an individual manipulate objects more efficiently).

Despite chimpanzee tool use possessing many signs of hierarchical organization, the majority of evidence for hierarchical organization in chimpanzee tool use comes from qualitative descriptions of behaviours. Descriptive arguments for behavioural organization have been pointed out on numerous occasions to be at risk of subjective bias. Even though sequences can be described hierarchically, this does not stop them from being generated using computationally simpler mechanisms of sequence generation (*Dawkins, 1976*; *Vereijken & Whiting, 1998*; *Girard-Buttoz et al., 2022*). This argument is reinforced by the observation that humans are biased to perceive hierarchical relationships in sequences which lack any form of hierarchical organization (*Fitch, 2014*; *Ferrigno et al., 2020*), which in this instance could include the observation of animals' behavioural sequences in the wild. There is consequently the need for new methodologies which are able to provide objective evidence that chimpanzee action sequences for tool use are organized hierarchically. Additionally, methodologies which also quantify the extent to which nonadjacent dependencies emerge between actions during such behavioural sequences would be particularly useful, as this is a property of chimpanzee tool use which has not yet been formally investigated.

To better characterise the sequential organization of actions during tool-use behaviours, and the extent to which such behaviours produce nonadjacent dependencies, we herein draw upon models of mutual information decay. Mutual information (MI) refers to the predictability of a given sequence element when the state of another element elsewhere in the sequence is known. MI often decreases between elements separated by greater distances; for example, looking out of a window at the weather one morning provides some information about the likely weather events that same afternoon. However, knowing the weather several days, or weeks prior would carry increasingly less information about the likely weather conditions in the upcoming afternoon. Whilst nonadjacent dependencies can be produced by a number of means during the generation of human sequential behaviours, several models have recently been developed that predict how MI decay with increasing sequential comparative distance, based on structural properties of sequences themselves (*Lin & Tegmark, 2017*; *Sainburg et al., 2019*; *Sainburg, Mai & Gentner, 2020*, *2022*; *Youngblood, 2024*). As non-hierarchical, 'regular' grammar systems (such as reflexive chains, and Markov models) produce sequences based on neighbouring probabilistic

dependencies, they exhibit exponential decreases in MI between elements separated by increasing sequential distances (*Lin & Tegmark, 2017*). Alternatively, sequences with hierarchical structures produce nonadjacent dependencies at greater distances. This is due to relationships between sequence elements imposed by higher-level aspects of a sequence's organization, and consequentially, hierarchical organization results in a power-law decay of dependencies over greater sequential distances (*Lin & Tegmark, 2017*; *Sainburg et al., 2019*; *Sainburg, Mai & Gentner, 2022*; *Youngblood, 2024*). Hierarchical and non-hierarchical structuring mechanisms may also be combined to form a composite structure (*Sainburg et al., 2019*; *Sainburg, Mai & Gentner, 2022*; *Youngblood, 2024*), where shorter, non-hierarchical strings are hierarchically organized to produce longer sequences (see above example of phonemes being organized into words, and words into sentences; *Sainburg et al., 2019*). Sequences produced by composite structuring may then display MI decay dynamics that are best approximated by exponential decay for sequential elements at local scales, and power-law decays for elements separated further across sequences. MI decay dynamics have been used to evaluate corpora of human and non-human vocal behaviours (*Sainburg et al., 2019*; *Sainburg, Mai & Gentner, 2022*; *Youngblood, 2024*), as well as the sequential actions of humans, zebrafish and fruit flies (*Sainburg, Mai & Gentner, 2020*). However, they have not thus far been applied to the sequential tool-use behaviours of a nonhuman animal.

In the present study, we employ MI decay models to investigate two overlapping research questions, concentrating on the sequential organization of serial action in chimpanzee tool use:

(1) Do chimpanzees produce nonadjacent dependencies within their sequential actions during tool-use behaviours, and if so, how to they compare with sequences from comparable Markov models?

(2) What do the dynamics of MI decay in chimpanzee tool-action sequences suggest about their sequential organization (Markovian, hierarchical, or a composite system)?

To answer these questions, we focus on the sequential organization of actions used by wild West-African chimpanzees (*Pan troglodytes verus*) when engaging in one of their most complex natural tool-use behaviours: the cracking of hard-shelled nuts with hammer and anvil stones. Through MI estimations, we demonstrate that the sequences of actions performed during tool use by two thirds of all sampled adult chimpanzees possess dependencies at significantly greater distances than those predicted by simulated Markov models. For half of adults, this relationship remained once repeating actions were accounted for, suggesting that repeating actions alone could not account for dependencies at long distances for many adults. We also report that the sequences of actions used by the majority of chimpanzees during nut cracking yield MI decay profiles that are concurrent with previous qualitative descriptions of hierarchical structuring in the tool action of apes. Additionally, we note that there is detectable interindividual variability in the distances at which dependencies occur in action sequences, and the most likely mechanisms of sequence generation detected by MI decay. Whilst our results are observational—and therefore limit the extent to which we can make claims about the underlying cognition of wild chimpanzees in this study—we report that the MI dynamics found in action

sequences for tool use support previous claims that chimpanzees organize these behaviours hierarchically; however, given interindividual differences, further research is required to understand if this is a universal system of behavioural organization possessed by chimpanzees. Thus, our findings provide additional evidence for the capacity of non-human apes to draw upon supraregular rules of sequence production, and produce nonadjacent dependencies during a natural behaviour: tool use. We discuss the need for further studies that will quantify the precise mental representations which permit chimpanzees to produce sequences that are organized across multiple temporal scales. In extension, our findings feed into the wider discussion surrounding the abilities of animals to produce complex sequential behaviours, including possibly homologous structural properties to those found in the behaviours of humans.

## METHODS

Portions of this manuscript were previously published as part of a preprint (https://doi. org/10.1101/2024.03.25.586385).

### Study site

Bossou-Nimba (07°390′N; 08°300′W) is one of a small number of long-term field sites focused on the study of wild West-African chimpanzees (*Matsuzawa, 2011*). Located in South-Eastern Guinea, Bossou is surrounded by primary and secondary forest, and is home to a small community of wild chimpanzees whose behaviour and ecology has been studied systematically since 1976. The community's home range covers an area of approx. 5–6 km (*Sugiyama, 1981*). Bossou chimpanzees are known to perform a wide variety of tool-use behaviours, including nut cracking (*Matsuzawa, 1994*; *Inoue-Nakamura & Matsuzawa, 1997*; *Biro et al., 2003*; *Carvalho et al., 2008*; *Hayashi, 2015*); ant-dipping (*Humle & Matsuzawa, 2002*), and pestle-pounding (*Yamakoshi & Sugiyama, 1995*).

The collection of video data of wild chimpanzees at Bossou has been conducted by many different researchers over several decades (*Matsuzawa, Hulme & Sugiyama, 2011*); these researchers collected data through non-invasive methods (see below), and prior to data collection, sought approval from the ethics committee at the Primate Research Institute, Kyoto University. Data used in the present study was collected under the joint approval of La Direction Nationale de la Recherche Scientifique et Technologique (DNRST), Guinea, and the Primate Research Institute, Kyoto University, Japan.

### Data collection

Data collection on nut-cracking behaviours is conducted at the "outdoor laboratory" at Bossou (*Matsuzawa, 1994*; see Fig. S1). Located in close proximity to the peak of Mount Gban, the outdoor laboratory is a natural rectangular clearing measuring approx. 7 × 20 m$^2$ and operating as an *in-situ* experimental facility during the dry season of each year (November–February). Chimpanzees visit the clearing spontaneously as part of their daily ranging (at a rate that varies from once or twice a day on consecutive days, to returning after several days), and are provided with a selection of numbered stones of varying sizes, shapes and raw-material types. Chimpanzees are also provided with several piles of nuts,

including naturally occurring oil-palm nuts (*Elaeis guineensis*), as well as Coula nuts, (*Coula edulis*), which despite not occurring naturally at Bossou, have been provided to chimpanzees at the outdoor lab since 1993 (*Biro et al., 2003*). A branch of oil-palm fruits is also provided, which is replenished frequently. Additionally, the outdoor laboratory features a tree trunk with a wide cavity inside, which acts as a water point. The outdoor laboratory is surrounded by thick vegetation on three sides, and on the fourth side an artificial wall made of palm fronds is used to allow human observers to observe chimpanzee behaviours out of sight, whilst simultaneously recording behaviours using tripod-mounted cameras. These conditions allow chimpanzee behaviours to be recorded in their natural habitat from close range but with minimal interference from human observers.

The several decades of video footage collected at Bossou have recently been compiled into a longitudinal video archive, spanning from 1988 to the present (*Schofield et al., 2019*). As such, the Bossou video archive presents a unique opportunity to study wild chimpanzee behaviour *in situ* through both longitudinal and cross-sectional lenses. This study takes a cross-section of video footage data from the Bossou video archive, focusing on nut-cracking behaviours recorded at the outdoor laboratory in the 2011/2012 field season.

## Subjects

Since the start of systematic research at Bossou, the site is known to have supported a community of around 18–22 individuals; however, the population has been in decline since a disease outbreak in 2003 (*Matsuzawa et al., 2004*), and as of October 2024, they number only three individuals. In the 2011/2012 field season, the community consisted of 10 adult individuals (four male and six female), one juvenile female and two infants males, with a community age range estimated to be from ~1 to ~54 years old (see Table S1). All available videos from the 2011/2012 field season were used during behaviour coding (see below), and all individuals who visibly interacted with a stone tool, nut, or nut fragment (including shells and kernels from their own or another's previous nut-cracking event) whilst present in the outdoor laboratory were included in behaviour coding.

## Behaviour coding

Behaviour coding was conducted using BORIS (Behavioural Observation Research Interactive Software; *Friard & Gamba, 2016*). Digitised video recordings were passed to BORIS and visualised on a 59 cm monitor. An ethogram of codable behaviours was developed based on *Inoue-Nakamura & Matsuzawa*'s *(1997)* nut cracking ethogram. Our ethogram included 34 manipulations performed by chimpanzees, *e.g.*, "Grasp", "Place", "Strike", "Peel" *etc*., (see Table S2 for full ethogram) and each manipulation was coded alongside one of a number of available objects, including: 1. Nut (uncracked), 2. Hammer, 3. Anvil, (Anvils were distinguished from Hammers on a case-by-case basis, based on how chimpanzees used each stone tool), 4. Kernel—taken to be the edible inner part of the nut, 5. Shell—taken to be the inedible outer parts of the nut, 6. Bare Hand. In our analysis, each of these bigram descriptions (*e.g.*, 'Grasp Nut') were considered to be an individual action. Some object-directed actions are continuous over time, and may occur simultaneously

alongside other behaviours, *e.g.*, a chimpanzee may support an anvil with its foot for an extended duration of time, whilst successively striking a nut with a hammer stone. To allow for analysis by mutual information decay, simultaneous actions were coded at the first point at which they were elicited; thus, all actions were coded as discrete-time points. This allows for actions to be described in one linear sequence (permitting MI decay analyses), whilst also retaining information surrounding the order in which chimpanzees began to externalize each object-directed action. Following action coding, each individual action was taken to constitute an individual sequence 'element' in our analysis (*i.e.*, "Grasp Nut" = 1 sequence element).

Action coding commenced when an individual began interacting with nuts, nut fragments or stone tools. Coding ceased when interacting with nuts and stone tools to engage in another behaviour. Individuals who ceased interacting with nuts, nut fragments, and stone tools for longer than 1 min, but were not clearly engaging in another behaviour, were deemed to have begun resting, and behaviour coding stopped. Behaviour coding also ceased if an individual engaging in nut-cracking behaviours could no longer be clearly observed from the video footage, such as through manoeuvring their body to block their behaviours from being visible to cameras, moving out of shot of the cameras, or cameras panning away from focal individuals. Under instances where chimpanzees moved back into view following termination of a sequence due to poor visibility, behaviour coding resumed. Thus, behaviour coding produced a mixture of complete sequences, (where individuals were observed engaging in tool use from start to end) as well as shorter fragments of the tool use sequence. These shorter fragments contained the actions used to crack a subset of nuts within a longer sequence of cracking nuts, as well as shorter sequences of actions for raw material acquisition, and the disposal of tools and inedible shell.

Prior to MI analyses, sequences were concatenated in the order they were collected. Additionally, for any action sequences which included successful cracking of nuts, we counted the number of actions used by an individual to crack open a nut, consume all corresponding kernel, and dispose of any waste shell. We then estimated the mean number of actions performed to crack individual nuts for each chimpanzee, for comparison with transition points (see *Estimating Transition Points in Composite Models*).

Behaviour coding was conducted exclusively by EHS. To ensure our ethogram produced repeatable results, we ran interobserver reliability tests using additional trained observers, which returned scores of 94–96% (see Supplemental Materials for further details).

## Mutual information estimation

To estimate MI values at various element distances, sequences for each individual were concatenated in order of collection. Mutual information can be determined by the reduction in entropy of a point in a sequence, when the state of another sequence element is known. As such, information can be estimated through independent measures of entropy at different points in a sequence (the marginal entropy of points X and Y), and also the joint entropy of actions co-occurring together at a given sequential distance. To estimate information in our action sequences, marginal and joint entropies were estimated
across concatenated sequences at distances ranging from 0 to 100 elements apart. MI was estimated at each distance as:

$$\widehat{MI}(X, Y) = \hat{S}(X) + \hat{S}(Y) - \hat{S}(X, Y),$$

where X is the distribution of the initial elements; Y is distribution of elements n-items further along the sequence; $\hat{S}(X)$ and $\hat{S}(Y)$ are the marginal entropies of X and Y respectively; $\hat{S}(X, Y)$ is the joint entropy of elements n distance apart, and $\widehat{MI}(X, Y)$ is the mutual information estimate for elements n distance apart. During entropy estimation, we followed the recommendations of *Sainburg et al. (2019)*, *Sainburg, Mai & Gentner (2020, 2022)* and *Lin & Tegmark (2017)* by using the Grassberger method (*Grassberger, 2003*), which accounts for under-estimation of true entropy from finite samples:

$$\hat{S} = log_2(N) - \frac{1}{N} \sum_{i=1}^{k} N_i \psi(N_i),$$

where $\hat{S}$ is the marginal or joint entropy; N is the total number of elements in the distribution; K is the number of different groups of elements in the distribution, and $\psi$ is the digamma function. To account for lower bounds of estimated MI, sequences underwent 1,000 pseudorandom permutations to generate randomised joint-entropy distributions. MI was then estimated using the mean of the permuted joint-entropy distribution, and the marginal entropy values from the in-sequence data, using the same equation as before. This MI estimate is taken to approximate MI measures which arise from our data due to chance relationships (*Sainburg, Mai & Gentner, 2020, 2022*). The chance MI measure—$\widehat{MI}_{sh}(X, Y)$—was subtracted from our data, to give an adjusted MI score which more accurately reflects the intentional structure of observed action sequences —$MI_{Adj}$:

$$\widehat{MI}_{sh}(X, Y) = \hat{S}(X) + \hat{S}(Y) - \hat{S}_{sh}(X, Y),$$

$$\widehat{MI}_{Adj} = \widehat{MI} - \widehat{MI}_{sh}$$

During MI estimation, it is possible for MI to decay to zero before reaching our maximal comparative distance of 100 elements. However, determining when $MI_{Adj}$ is zero is challenging, due to the fact that it is calculated by subtracting one estimate from another, and is therefore unlikely to be precisely zero. Rather, when $MI_{Adj}$ was truly zero, it would randomly fluctuate about this value, driven by small-scale stochasticity in the estimates of MI and $MI_{sh}$. Considering this fact, we used the 1,000 sequence permutations we previously generated to construct 95% confidence intervals surrounding $\widehat{MI}_{sh}(X, Y)$ at each sequential distance between 0–100 elements. At each sequential distance between 1–100 elements, we took the upper value of the 95% confidence limit, and subtracted this value from the $\widehat{MI}$ estimated from the original action-sequences at each corresponding distance:

$$\widehat{MI} - CI_{95\%}^{Upper} \left\{ \widehat{MI}_{sh}(X, Y) \right\}$$

We took the first comparative distance this value fell below zero to indicate that estimated MI was no longer significantly different to what could be expected from chance relationships, and therefore could no longer be treated as different to zero. Since we used the upper value of the 95% confidence interval, instead of the mean value, this is a conservative estimate of this maximum distance for which MI was positive in analysed sequences. For MI decay profiles which decayed to zero before reaching the maximum comparative distance of 100 elements, MI decay data was restricted to include solely the comparative distances that preceded the complete decay of MI. For all subsequent analyses and modelling, the adjusted MI score was used ($MI_{Adj}$).

## MI decay model fitting and selection

Three models were used to characterise decay of adjusted MI scores: (1) an exponential model to approximate $MI_{Adj}$ decay dynamics predicted by sequences produced by Markovian, adjacent dependencies (*Lin & Tegmark, 2017*; *Sainburg et al., 2019*; *Sainburg, Mai & Gentner, 2022*); (2) a power-law model to approximate $MI_{Adj}$ decay predicted by fully-nested hierarchical sequences (*Sainburg et al., 2019*; *Sainburg, Mai & Gentner, 2022*); and (3) a composite model to approximate $MI_{Adj}$ decay predicted by sequences which utilise nested hierarchical structuring at higher orders of abstraction, before ordering elements through Markovian dynamics at local scales (see *Sainburg et al., 2019*; *Sainburg, Mai & Gentner, 2022* for further examples of composite MI decay in human and animal behaviours):

$$Exponential\ Decay = a * e^{-x*b}$$

$$Power\text{-}Law\ Decay = a * x^b$$

$$Composite\ Decay = a * e^{-x*b} + c * x^d$$

where x represents inter-element distance, and a, b and c are coefficients. As reducible and periodic Markov chains may produce $MI_{Adj}$ scores which decay to positive constants (*Lin & Tegmark, 2017*), best-estimate Markov chains were modelled from sequence data for each individual, and checked to determine whether they were irreducible and aperiodic. Markov models were estimated in R using the *markovchainFit()* function from the *markovchain* package (*Spedicato, 2017*). Irreducibility and periodicity of these Markov models were then determined using the functions *is.irreducible()* and *period()* respectively. As all best-estimate Markov chains were identified as irreducible and aperiodic, we fitted our exponential model in the absence of a constant, allowing models of $MI_{Adj}$ to decay towards zero. Both the power-law and composite decay models were also fitted in the absence of constants to also allow these models of $MI_{Adj}$ to fall towards zero with increasing sequential distances (see the *Statistical Analysis* section for more information on model fitting).

The candidate MI decay models (exponential, power-law and composite) were fit to $MI_{Adj}$ data using the nlsLM() function of the minpack.lm package (*Elzhov et al., 2016*), which uses the Levenberg-Marquardt algorithm to fit non-linear curves by least-squares.

The fit of the three models were compared using an adaptation of the Akaike Information Criterion (AIC). AIC provides a numerical means to compare goodness-of-fit between models constructed on identical data-sets, whilst also penalising extra parameters used to construct the model. Such penalization of additional model parameters thus helps to reduce the likelihood of overfitting more complex models to their underlying data. Models with lower AIC scores are taken to give more accurate approximations of data, relative to the number of parameters used to fit them. To mirror similar analyses of MI decay on birdsong, language, and human action (*Sainburg et al., 2019*; *Sainburg, Mai & Gentner, 2020*, *2022*), we opted to compare model fits using AICc. AICc scores models similarly to AIC; however, AICc introduces an additional penalty for each model parameter, to correct for overfitting on smaller datasets (*Cavanaugh & Neath, 2019*; *Sainburg et al., 2019*; *Sainburg, Mai & Gentner, 2020*, *2022*).

As repeating elements in a sequence (*e.g.*, successively striking a hammer) can produce nonadjacent sequential dependencies, we repeated the analyses outlined above—specifically in the sections *Mutual Information Estimation* and *M. Decay Model Fitting and Selection*—on a corpus of condensed sequences for each individual. Condensed sequences were produced by passing the concatenated sequence corpus for each individual through an algorithm which removed repeated actions are replaced them with a single codon (*e.g.*, 'Strike Hammer', 'Strike Hammer', 'Strike Hammer', is replaced by a single codon: 'Strike Hammer). By analysing this condensed corpus of sequences in parallel with observed sequences of chimpanzee tool action, we are able to separate out dependencies produced by repeated actions, and those produced by alternative means, including the hypothesised use of nested subroutines.

## Estimating transition points in composite models

For those individuals whose $MI_{Adj}$ decay profiles were best approximated by a composite model, we identified the comparative scale at which the model transitioned from exponential decay to power-law decay using the change in gradient of the curve in log-space ($\log_{10} MI_{Adj}$ and $\log_{10}$ element distance). Through finding the second differential of the curve in log-space, we took the minimand of this function to be the point immediately prior to where the function begins to switch to a power-law decay. This is the greatest inter-element distance where comparisons are likely occurring between elements in the same subroutine (which the composite model predicts are structured as serial chains). At greater comparative distances, elements begin to be more commonly compared between subroutines (which the composite model predicts are structured hierarchically). We can therefore use the transition point for composite models to provide an approximate prediction of subroutine length within action sequences.

## Markov models and simulated sequences

Sequences produced from some Markovian grammars may produce dependencies which span over multiple elements (including short-range nonadjacent elements), especially when Markov models have highly predictable element transitions, leading to multiple elements being produced in highly stereotyped strings. It is therefore necessary to

determine whether the maximum distances over which dependencies occur in chimpanzee tool-action sequences are significantly greater to those that can be readily generated by equivalent Markov models. Using the Markov models generated from observed action sequence data from individual chimpanzees (see section above: *M. Decay Model Fitting and Selection*), we generated 100 Markov sequences matching the total corpus length for each individual (see Fig. S3 for a graphical illustration of these Markov models). For all Markov sequences, we estimated $MI_{Adj}$ Decay using the same method as outlined in the sections *Mutual Information Estimation* and *M. Decay Model Fitting and Selection*. From the 100 Markov Sequences for each individual, we constructed a probability distribution for the maximum sequential distances where dependencies could be detected, using the maximum sequential distance $MI_{Adj} > 0$ for each sequence. From these probability distributions, 95% confidence intervals were constructed using the exact method for Poisson distributions. To determine whether dependencies occurred at greater distances in observed action sequences compared to those expected from Markovian structuring, we compared the maximum sequential distances where $MI_{Adj} > 0$ in observed action sequences (a single point estimate for each chimpanzee) *vs.* to their corresponding 95% confidence interval for Markov sequences. To assess whether the maximum distances over which dependencies were detected was influenced by repeating actions, this process was repeated for each individual using condensed sequence data, including retraining Markov models using condensed action sequences.

To determine the extent to which Markovian approximations of chimpanzee tool-action produce $MI_{Adj}$ decay profiles which are best approximated by exponential decay dynamics, we used two 'community' Markov models to produce control sequences. We concatenated sequence data from all eight individuals, and used this extended corpus to train an initial 'community' Markov model. This community Markov model reflected all possible sequential-action combinations seen by all individuals within our analysis (similar to E-Languages in linguistics, which reflect the corpus of externalized utterances within a population; *Chomsky, 1986*). From this community Markov model, we simulated 500 sequences, each 1,000 elements long, which were used for $MI_{Adj}$ Decay analyses and Model fitting as outlined in sections *Mutual Information Estimation* and *M. Decay Model Fitting and Selection*. The proportion of MI decay profiles which were best characterized by exponential decay dynamics were calculated using AICc scores. This process was repeated with condensed sequence data, to produce a second community Markov model which approximated sequential relationships in the absence of repeating actions (see Fig. S5 for a graphical illustration of both community Markov models). For MI decay analyses from Markovian sequences, the University of Oxford's Advanced Research Computing Facility was used to run MI Decay analyses in parallel (*Richards, 2015*).

# RESULTS

## Sampled sequence data

We analysed a total of 251.85 min of video data of chimpanzees engaging in nut cracking, yielding action sequences from nine individuals (see Table 1; see Table S1 for demographic data of sampled chimpanzees). Of these nine individuals, one (Velu) had fewer than 200

**Table 1 Summary of sequence data collected from the 2011/2012 field season at Bossou (including all complete and incomplete sequences).**

| Individual | Sex | Age (years) | Sampled for MI analysis | Time coded (MM:SS) | Number of sequences | Sequence elements (Total) | Mean sequence length ± SE | Number of action types | Nuts cracked | Mean sequence length per nut ± SE |
|---|---|---|---|---|---|---|---|---|---|---|
| Fanle | F | 14 | ✓ | 44:55 | 21 | 1,368 | 65.1 ± 1.8 | 34 | 85 | 13.9 ± 0.4 |
| Flanle | M | 4 | ✓ | 24:03 | 35 | 453 | 12.9 ± 0.6 | 59 | 1 | 48 |
| Foaf | M | 31 | ✓ | 07:22 | 8 | 299 | 37.4 ± 2.2 | 18 | 10 | 31.2 ± 1.8 |
| Jeje | M | 14 | ✓ | 40:33 | 18 | 1,373 | 76.3 ± 2.1 | 36 | 39 | 29.4 ± 0.9 |
| Jire | F | 53* | ✓ | 19:27 | 12 | 482 | 40.2 ± 1.8 | 33 | 22 | 16.0 ± 0.9 |
| Joya | F | 7 | ✓ | 36:52 | 39 | 1,178 | 30.2 ± 0.9 | 55 | 22 | 36 ± 1.3 |
| Peley | M | 13 | ✓ | 34:05 | 15 | 1,168 | 77.9 ± 2.3 | 44 | 45 | 21.7 ± 0.7 |
| Tua | M | 54* | ✓ | 55:33 | 15 | 1,940 | 129.3 ± 2.9 | 43 | 79 | 22.4 ± 0.5 |
| Velu | F | 52* | ✗ | 01:07 | 2 | 16 | 8 ± 2 | 12 | 0 | 0 |

Note:
Eight individuals were included in MI decay modelling. 'Time Coded' relates to the total duration of time individuals were seen to be engaging with nuts, nut fragments and stone tools throughout all video footage. An 'Element' is defined as each action coded within a sequence, whereas 'Action Types' represent the different types of coded actions in a sequence (e.g., "Grasp Nut" "Place Nut" "Grasp Nut"—Number of elements: 3, Number of Action Types: 2). 'Nuts Cracked' marks the number of nuts observed to have been cracked by an individual across all available video footage. Mean Sequence Length Per Nut refers to the mean number of elements in a sequence that is directed at cracking open each individual nut. SE for all means were calculated via the normal approximation of the Poisson distribution. Ages marked with * are estimates, as these individuals were already present at Bossou when long-term monitoring began in 1976.

actions coded throughout the entire series of sampled videos. As entropy estimation from small sample sizes can be subject to high levels of error (*Grassberger, 2003*), this individual was omitted from our MI analyses. Subsequently, eight individuals were included in our MI analyses: two adult females (Fanle & Jire); four adult males (Foaf, Jeje, Peley & Tua); one juvenile female (Joya) and one infant male (Flanle).

Across the eight individuals in our dataset, we recorded action sequences for the cracking of 303 individual nuts, using 8,261 individual actions, belonging to 82 unique classes of action (see Fig. 1 for a list of the most common classes of actions used by chimpanzees, and example sequences of actions performed during nut cracking). Flanle, the infant male, used the greatest variety of different actions during nut cracking (59 unique classes of action), followed by Joya, the juvenile female (55 unique classes of action). Both of these individuals interspersed short sequences of solo object play into their attempts at nut cracking, e.g., mouthing the hammer stone, coded as either 'kiss HAMMER' or 'bite HAMMER', depending on the force applied. For both of these individuals, we retained these actions as part of our coded action sequences, to avoid subjective bias of when individuals were or were not attempting to crack nuts.

Adults used a smaller repertoire of actions when engaging in nut cracking. This repertoire included a small subset of high-frequency actions to facilitate nut cracking, with the five most common actions including 'Strikeonehand Hammer' (24.6%), 'Eat Kernel' (11.8%), 'Grasp Nut' (10.5%), 'Place Nut' (8.6%), and 'Grasp Kernel' (8%;). In total, 63% of all actions within our dataset fell into one of these five action types. In addition to these frequently performed actions, an extended number of actions were performed less frequently, only where necessary, e.g., passing the hammer between the hands (pass HAMMER), or rotating the anvil stone to confer stability (reorient ANVIL; see Fig. 1C for more examples).

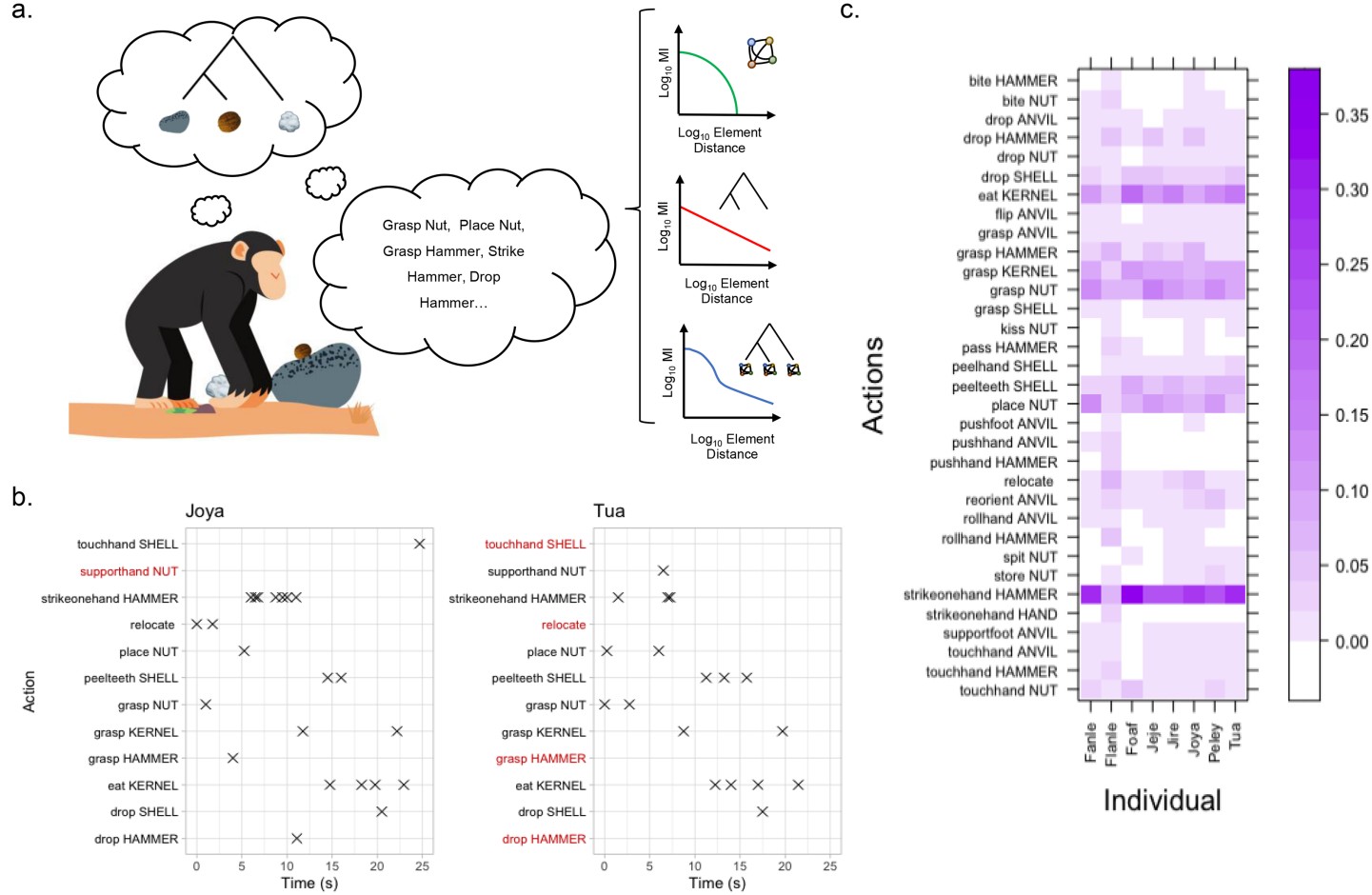

**Figure 1 MI decay models and the diversity of tool-use actions employed by chimpanzees during nut cracking.** (A) A chimpanzee using two stone tools to crack open a nut (tree structure diagram represents hierarchical association of objects—see *Matsuzawa, 1996*), and a complementary sequence of actions to facilitate object associations. To the right of the action sequence, three candidate structuring mechanisms are outlined, alongside their characteristic MI decay profiles (from top to bottom: purely Markovian, purely hierarchical, a composite of a hierarchical structure with Markov models at the terminal nodes). Images for this figure were acquired through canva.com. (B) Two representative behaviour-time plots showing the sequence of actions used by Joya (left) and Tua (right) to crack and consume a nut. Crosses (×) mark where an action is coded as a discrete time point. Red action labels indicate where an action-type was not used by an individual when cracking and consuming each specific nut; however, they do not preclude these actions from being used by either individual when processing other nuts, and do not represent all possible actions. (C) A heatmap showing the proportion of different action types in the total corpus of action-sequence data for each individual (restricted to action types which constituted at least 1% of total observed actions for at least one individual). The darker purple shading means action types constitute a greater proportion of the corpus for each individual. An enlarged version of this figure can be found in the Supplemental Materials.

## Maximum inter-element distance MI detected

To understand the most extreme extent to which chimpanzees produced nonadjacent dependencies in their sequential actions, we identified the maximum sequential distance at which dependencies occurred. For each individual, we compared this result to the maximum distances $MI_{Adj}$ was positive in 100 sequences simulated from Markov models. This allowed us to determine whether dependencies in observed action sequences occurred at greater distances than those which can be produced through Markovian chaining of actions.

**Table 2 Summary of maximum comparative distances, model preferences, and transition points for action sequences with repeating actions.**

| Individual | Sex | Age class | Maximum sequential distance $MI_{Adj} > 0$ | Preferred model | Transition point | Mean sequence length per nut ± SE (no. of nuts cracked) |
|---|---|---|---|---|---|---|
| Fanle | F | Adult | 17 | Power-Law | – | 13.9 ± 0.4 (85) |
| Flanle | M | Infant | 100 | Composite | 2 | 48 (1) |
| Foaf | M | Adult | 9 | Exponential | – | 31.2 ± 1.8 (10) |
| Jeje | M | Adult | 22 | Composite | 4 | 29.4 ± 0.9 (39) |
| Jire | F | Adult | 11 | Power-Law | – | 16.0 ± 0.9 (22) |
| Joya | F | Juvenile | 21 | Composite | 8 | 36 ± 1.3 (22) |
| Peley | M | Adult | 42 | Composite | 2 | 21.7 ± 0.7 (45) |
| Tua | M | Adult | 20 | Composite | 7 | 22.4 ± 0.5 (79) |

**Note:**
Transition point describes the element distance where the composite model switches from a primarily exponential decay dynamic to one which is better generalised by a power-law. For each individual, the transition point, and mean number of actions used to crack open individual nuts, is provided. Total sample size for nut sequence length (*i.e.*, number of nuts cracked throughout the study period) is included in bracket.

On average, the maximum sequential distances where $MI_{Adj}$ remained above zero was 30.3 elements ($N = 8$, SD = 29.9; see Table 2). For Flanle, the infant male, $MI_{Adj}$ did not fall to zero before the maximum distance of 100 sequence elements. When excluding Flanle from this average—for whom the maximum distance over which dependencies were detected was over two standard deviations higher than all other individuals—the mean maximum sequential distance at which $MI_{Adj}$ was positive was 20.3 elements ($N = 7$, SD = 10.8).

For six individuals (Fanle, Flanle, Jeje, Joya, Peley and Tua), a positive $MI_{Adj}$ was detected at sequential distances that were significantly greater than those found in counterpart Markovian sequences (identified by a higher maximum inter-element distance where $MI_{Adj}$ was positive compared to the upper 95% CI of the corresponding Markov distribution; see Fig. 2 and Table S3). This result confirms that nonadjacent dependencies for these individuals occurred at distances greater than could be readily generated by comparative Markov models. Alternatively for two individuals, (Foaf & Jire), we found no difference between the maximum distance $MI_{Adj}$ was detected in the observed action sequences compared to sequences simulated from Markov models.

To determine whether successive repeats of elements (*e.g.*, repeatedly striking a hammer) influenced the maximum inter-element distance where $MI_{Adj}$ was above zero, observed action sequences were passed through a condensing algorithm that removed repeated elements, and $MI_{Adj}$ was estimated in the same way as before. Separate Markov models were also constructed from condensed data, and our analysis was repeated identically to before.

For the condensed action sequences, the mean maximum sequential distances where $MI_{Adj}$ was detected was 13.3 elements ($N = 8$, SD = 5.9). Flanle, the infant male, showed the largest decrease in the maximum distance where $MI_{Adj}$ was greater than zero, reducing from possibly over 100 elements down to nine sequence elements. This implies that dependencies were heavily influenced by repeating actions in Flanle's action sequences, possibly through repeated play actions.
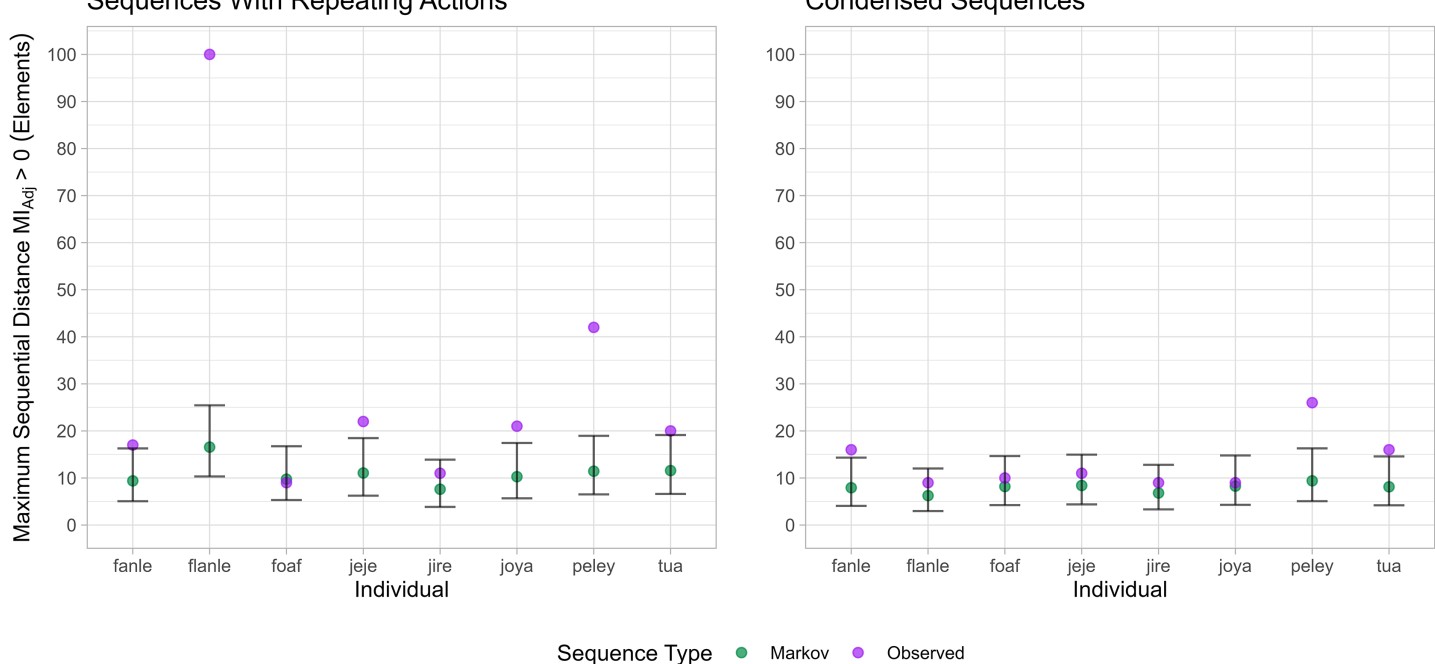

**Figure 2** **The maximum sequential distance dependencies were detected in action sequences and sequences from Markov models.** Determined by the maximum distance that $MI_{Adj}$ was estimated to be greater than zero. $MI_{Adj}$ estimated from action sequences is in (purple), and estimates from sequences generated by an individual's corresponding Markov models (green). The panel on the left is for data which includes repeating actions; the panel on the right corresponds to condensed action-sequence data. Bars represent a 95% confidence interval around estimated means for Markovian sequence data.

For three individuals (Fanle, Peley and Tua), the maximum distance at which $MI_{Adi}$ was positive remained significantly greater than sequences produced by their corresponding Markov models (see Fig. 2; see Table S3). These three individuals represent half of all adults in our sample. Following sequence compression, three individuals no longer exhibited dependencies at sequential distances greater than simulated Markov sequences: Flanle (infant male), Joya (juvenile female) and Jeje (adult male).

## MI decay analysis: observed action sequences

To understand how sequences of behaviours are generated, it can be informative to assess how mutual information decays between sequence elements which are progressively further apart (*Lin & Tegmark, 2017*; *Sainburg et al., 2019*; *Sainburg, Mai & Gentner, 2020*, *2022*; *Youngblood, 2024*). For seven out of eight individuals, $MI_{Adj}$ decay was best characterised by a model which included a power-law decay dynamic, which is a key indicator of hierarchical organization of elements across biological sequences (*Sainburg et al., 2019*; *Sainburg, Mai & Gentner, 2020*, *2022*; *Youngblood, 2024*; see Fig. 3 and Table 2; see Table S4 for AICc values). The $MI_{Adj}$ decay profiles of two individuals (Fanle and Jire) were best explained by a purely power-law model (predicted by solely hierarchical structuring); while for five individuals (Flanle, Jeje, Joya, Peley and Tua), the optimal decay model was identified as the composite model, which combines periods of exponential and power-law decay. These composite models follow decay profiles predicted by structuring

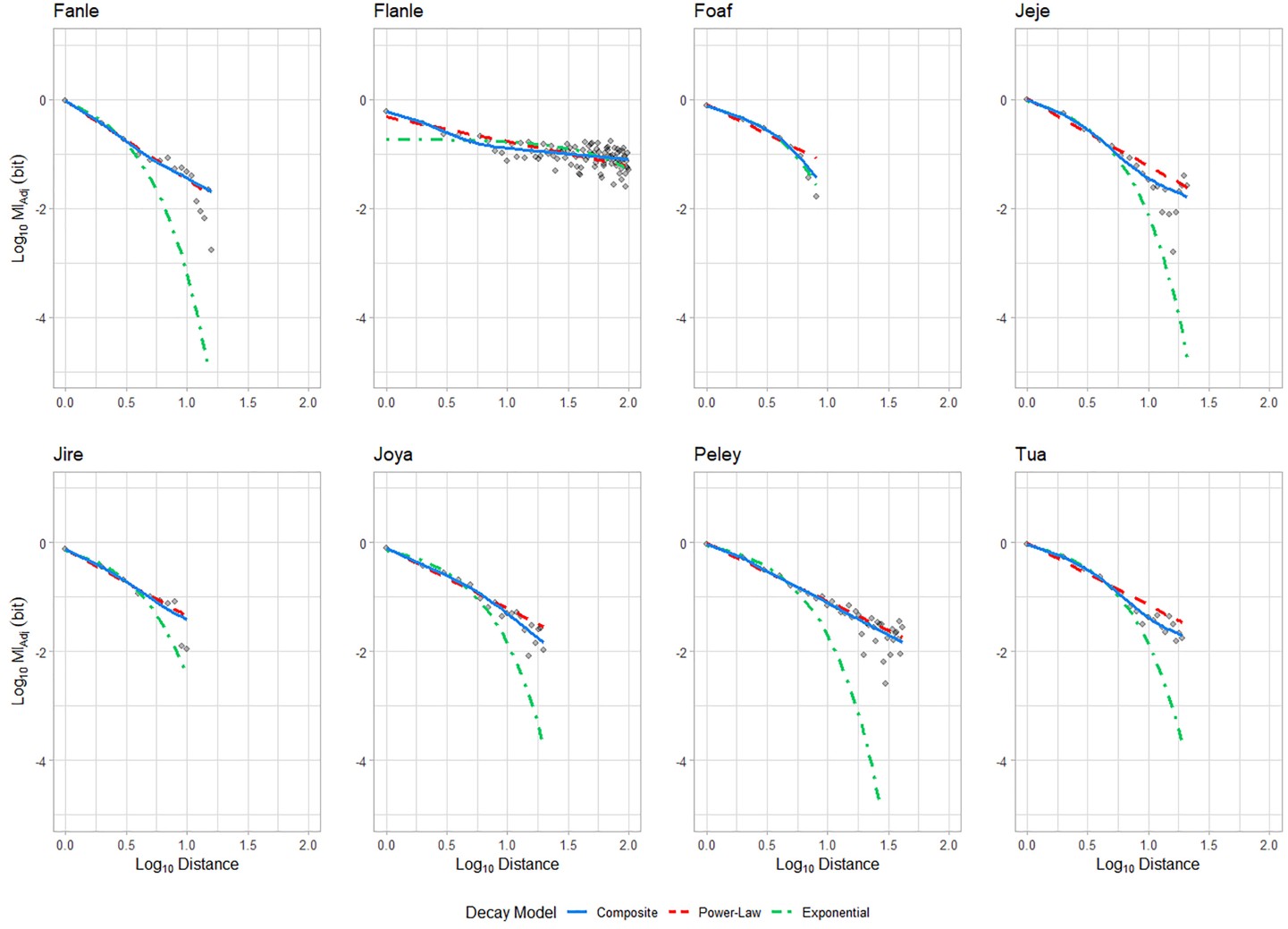

**Figure 3** $MI_{Adj}$ **decay profiles for each individual.** The three candidate decay models are fitted. The exponential model is in green (dot-dash); the power-law model is in red (dashed), and the composite model is in blue (solid line).

mechanisms which involve concatenation of elements into short subroutines based on adjacent transition probabilities, which are themselves hierarchically organized into longer sequences. In the case of these five adult chimpanzees, this could be reflected by a two-stage process of generating short subroutines through chaining together a handful of actions, before hierarchically organizing these subroutines into longer sequences. For one individual (Foaf), the exponential model was identified as the optimal model, implying that Foaf organized his actions using non-hierarchical means.

Similarly to our previous analysis of the maximum inter-element distances where we detected positive $MI_{Adj}$, we repeated the analysis above on condensed sequences to ensure power-law relationships were not simply a product of highly repetitive elements (see Table 3). When analysing condensed action sequences, we were able to identify an optimal decay model for the action sequences of six individuals. Of the eight individuals in our

**Table 3  AICc scores for MI$_{Adj}$ decay models, derived from condensed sequences.**

| Individual | Max element distance | Model | K | AICc | ΔAICc | AICc.Wt | Cum.Wt | LL |
|---|---|---|---|---|---|---|---|---|
| Fanle | 16 | **Composite** | **5** | **−70.5** | **0.0** | **0.93** | **0.93** | **43.6** |
| | | Power-Law | 3 | −65.2 | 5.3 | 0.07 | 1.00 | 36.7 |
| | | Exponential | 3 | −47.2 | 23.3 | 0.00 | 1.00 | 27.7 |
| Flanle | 9 | **Power-Law** | **3** | **−18.5** | **0.0** | **0.98** | **0.98** | **15.3** |
| | | Exponential | 3 | −10.3 | 8.2 | 0.02 | 1.00 | 11.2 |
| | | Composite | 5 | 0.8 | 19.4 | 0.00 | 1.00 | 19.6 |
| Foaf | 10 | **Exponential** | **5** | **−36.7** | **0.0** | **1.00** | **1.00** | **23.8** |
| | | Power-Law | 3 | −22.7 | 14.0 | 0.00 | 1.00 | 16.8 |
| | | Composite | 3 | −14.1 | 22.6 | 0.00 | 1.00 | 22.1 |
| Jeje | 11 | **Composite** | **5** | **−43.0** | **0.0** | **0.72** | **0.72** | **34.0** |
| | | **Exponential** | **3** | **−41.2** | **1.8** | **0.28** | **1.00** | **25.6** |
| | | Power-Law | 3 | −23.0 | 20.0 | 0.00 | 0.00 | 16.5 |
| Jire | 9 | **Power-Law** | **3** | **−22.7** | **0.0** | **0.54** | **0.54** | **17.4** |
| | | **Exponential** | **3** | **−22.4** | **0.3** | **0.46** | **1.00** | **17.2** |
| | | Composite | 5 | 10.7 | 12.0 | 0.00 | 1.00 | 25.4 |
| Joya | 9 | **Power-Law** | **3** | **−26.5** | **0.0** | **1.00** | **1.00** | **19.2** |
| | | Exponential | 3 | −15.0 | 11.5 | 0.00 | 1.00 | 13.5 |
| | | Composite | 5 | −2.3 | 24.2 | 0.00 | 1.00 | 21.2 |
| Peley | 26 | **Composite** | **5** | **−121.8** | **0.0** | **0.80** | **0.80** | **67.5** |
| | | Power-Law | 3 | −119.1 | 2.7 | 0.20 | 1.00 | 63.1 |
| | | Exponential | 3 | −80.0 | 41.8 | 0.00 | 1.00 | 43.6 |
| Tua | 16 | **Composite** | **5** | **−92.2** | **0.0** | **1.00** | **1.00** | **54.5** |
| | | Exponential | 3 | −70.7 | 21.5 | 0.00 | 1.00 | 39.4 |
| | | Power-Law | 3 | −49.0 | 43.2 | 0.00 | 1.00 | 28.6 |

**Note:**
For each individual, Max Element Distance highlights the greatest element distance included in the MI$_{Adj}$ datasets, and models are given in the order of preference (as determined by AICc—rows in bold indicate highest preferences; lower AICc values reflect better performing models). K represents the number of parameters estimated in each model. ΔAICc represents the difference between a given model's AICc score and the optimal model's AICc score for each individual. AICc.Wt describes the proportion of the total predictive power of all three models contained in each model, and Cum.Wt describes the accumulation of predictive power in order of model preference. LL represents the log-likelihood of each model given the data—AICc is estimated using both K and LL.

analysis, the action sequences of five individuals were best explained by a model which included a period of power-law decay (two by a purely power-law, and three by a composite model), predicted by composite or wholly hierarchical structuring. For two individuals (Jeje and Jire), multiple models offered competing explanations of decay dynamics: for action sequence data collected from Jeje, the composite and exponential models offered competing explanations of the data; whereas for Jire, the power-law and exponential models exhibited similar explanatory power. For these individuals, we therefore could not find conclusive support for any one structuring mechanism. For the MI$_{Adj}$ decay profile for Foaf, the exponential model continued to offer the best explanation

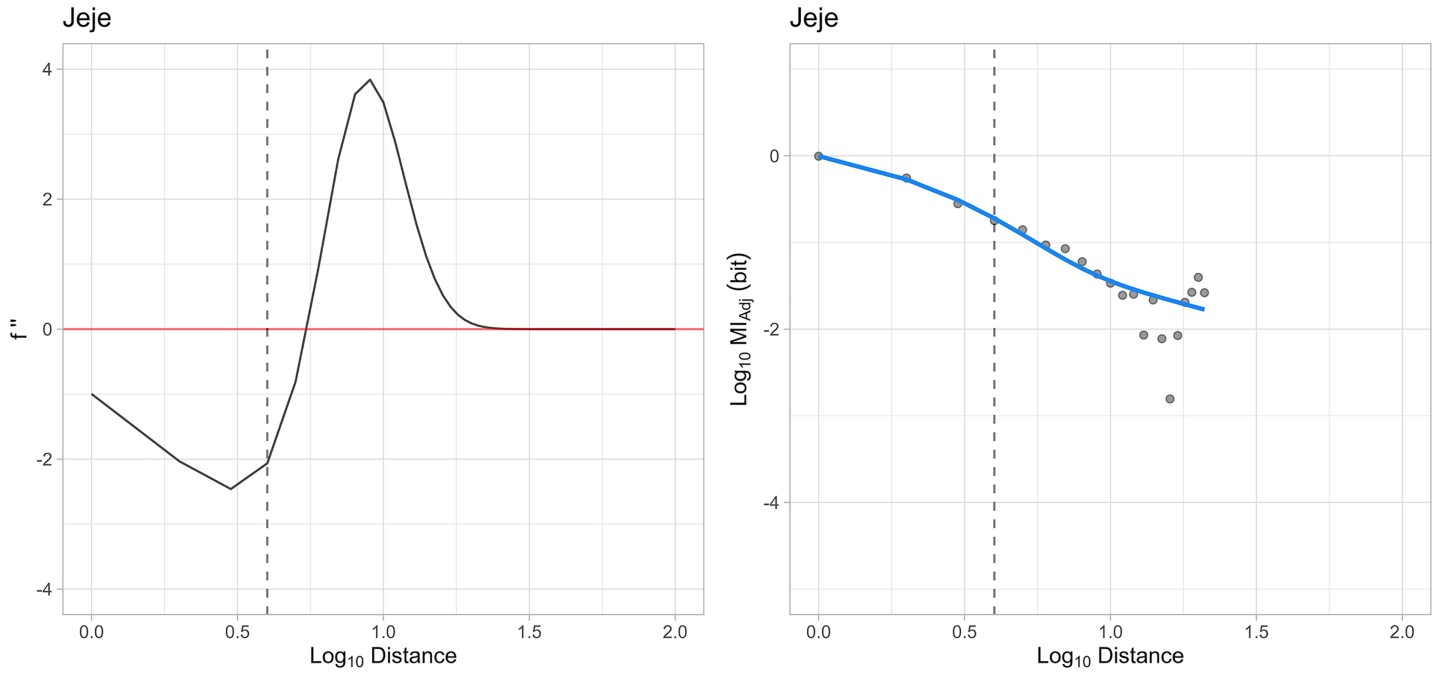

**Figure 4 Example curvature analysis for the composite model.** The panel on the right shows the composite model describing $MI_{Adj}$ decay for Jeje. The panel on the left illustrates the second differential of this composite function. The point after the minimand of the second differential indicates the transition to power-law decay (indicated by a vertical dashed line). This point is mirrored on the plot on the right by the same line.

of the data following sequence compression, implying non-hierarchical means of action-sequence generation.

## Transition points for composite models

For each individual for whom our analysis showed a preference for the composite decay model, we also identified the transition point between primarily exponential and primarily power-law decay dynamics using the second differential of the curve in log-space (see Fig. 4.; see Methods section *Estimating Transition Points in Composite Model*). The maximum comparative distance prior to the transition between exponential and power-law decay dynamics ranged from 2–8 elements (mean = 4.6 elements, $N = 5$; see Table 2; see Fig. S4 for transition point profiles for each individual whose $MI_{Adj}$ is best explained by the composite model). This predicts that for individuals whose sequence structure is best approximated by composite models, non-hierarchical chaining of actions can be used to produce subroutines of 2–8 elements apart. In extension, the composite model predicts that these subroutines are combined hierarchically to produce extended behavioural sequences. Additionally, for all of these individuals, transition points were identified at distances which were significantly shorter than the sequences of actions used to crack open individual nuts (see Table 2). This result reaffirms that composite decay dynamics were not produced through concatenation of action sequences directed at individual nuts, but were likely produced through sequential structuring of actions by chimpanzees at much shorter timescales.

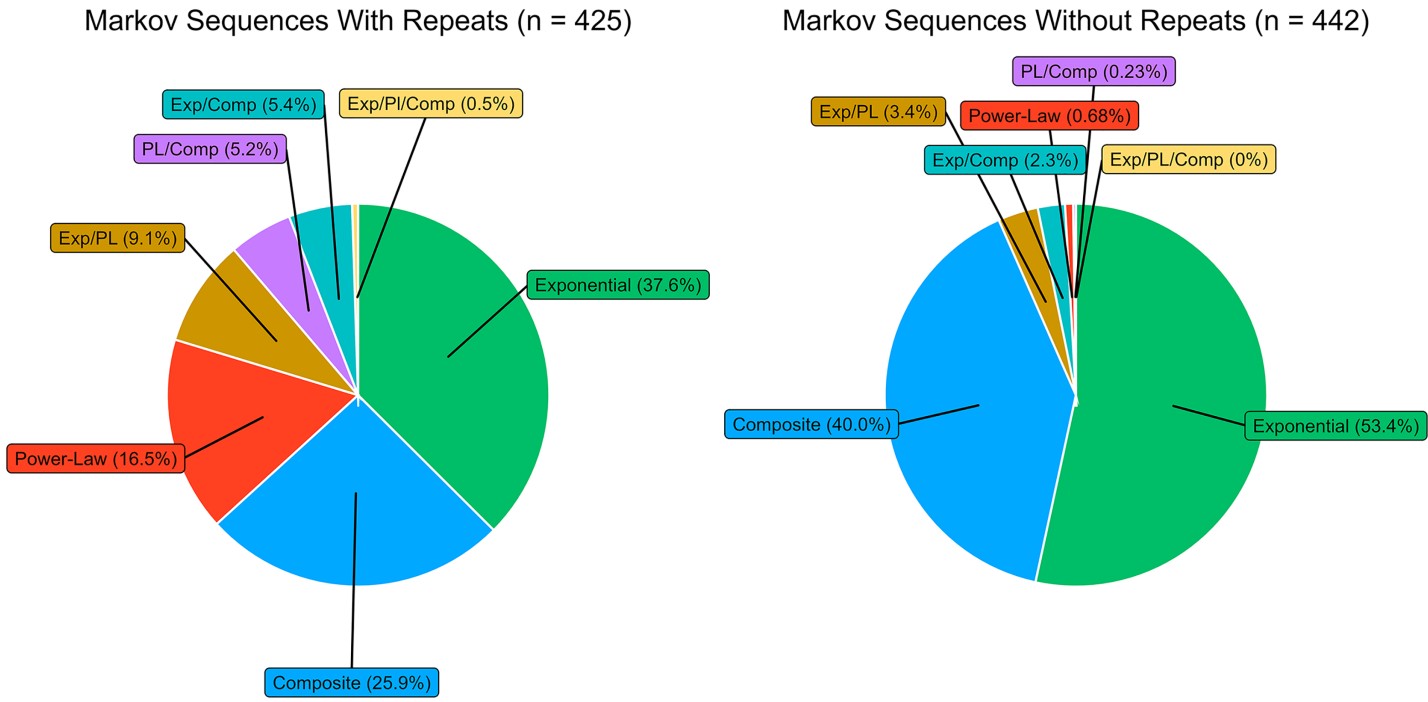

**Figure 5** The proportion of instances MI$_{Adj}$ decay was preferred when generating sequences from community Markov models. The panel on the left relates to 425 sequences generated from a Markov model which can produce repeating actions in a sequence. The panel on the right relates to 442 sequences generated from a Markov model which cannot produce repeating actions.

## MI decay analysis: Markovian sequences

To understand the rate at which our analyses of MI decay would identify power-law decay dynamics in non-hierarchical sequences (akin to a false-positive rate for identifying signatures of hierarchies from non-hierarchical sequences), we simulated action sequences from community Markov models, and ran the same model selection process as conducted on chimpanzee action sequences. We estimated a false positive rate by identifying the proportion of Markovian sequences whose MI decay profiles were best described by models which contained power-law relationships.

For the Markov model trained on the entire corpus of action-sequence data, including repeating elements, all three decay models were successfully fitted for the MI$_{Adj}$ decay profiles of 425 sequences. There was a significant, positive relationship between with the maximum sequential distance at which MI$_{Adj}$ was detected, and the likelihood of successfully fitting all three models (Binomial GLM with Logit Link: Success ~ Maximum Sequence Length, $N = 425$; Intercept: $-9.25$; Maximum Length: 1.53; $Z = 8.96$; $p < 0.001$), and this relationship also held for the condensed data (see Table S5). Of the 425 model comparisons, AICc identified a single best model for characterising MI$_{Adj}$ decay for 340 datasets (80% of all model comparisons). 52.6% of model comparisons identified the exponential model as highly competitive (see Fig. 5). In 37.6% of model comparisons the exponential model was identified as the sole best performing model, and in 15% of comparisons the exponential model was identified as one of multiple equally optimal

models. 16.5% of comparisons showed a sole preference for the power-law model, and 25.9% of comparisons showed a sole preference for the composite model.

This rate suggests that, if chimpanzees were organizing sequences similar to non-hierarchical Markov models, we would expect composite and power-law models to be the sole best performing model in 42.4% of instances combined (approximately three of eight individuals in total). In the action sequence data, this value was over double (seven individuals; composite = five individuals; power-law = two individuals). Our simulation also predicts that, if generated through non-hierarchical means, the exponential model would be the best performing model for three individuals. However, our analysis of the action-sequence data revealed that the exponential model was preferred for a single individual, Foaf.

A similar result was found when using a Markov model trained on condensed action sequences (see Fig. 5). If action sequences were generated through non-hierarchical means, we would expect composite and power-law models to be the sole best performing model in 40.68% of instances (three of eight individuals). We identified five individuals for whom composite and power-law models were the sole preferred model (composite = three individuals; power-law = two individuals). Equally, we would expect the exponential model to be preferred in 53.4% of instances (four individuals); however, much like the analysis for action-sequence data containing repeating actions, the exponential model was preferred for a single individual, Foaf.

## DISCUSSION

Nonadjacent dependencies emerge during the performance of many elaborate sequential behaviours in humans, and can be the result of hierarchical organization (*Lashley, 1951*; *Dawkins, 1976*; *Mel'čuk, 1988*; *Chomsky, 2002*; *Rosenbaum et al., 2007*; *Fitch & Martins, 2014*; *Rispens & Soto de Amesti, 2017*; *Wilson et al., 2020*). However, these cognitively demanding aspects of sequential behavioural organization appear to be sporadically distributed across other animal taxa, and when present, may be restricted to specific behaviours. Given their close phylogenetic proximity to humans, characterizing how great-apes organize sequential behaviours is informative for understanding how human sequential behaviours evolved. The purpose of this study was to apply mutual information measures (estimates of dependency strength between sequence elements) to: (1) determine whether chimpanzees routinely produce nonadjacent dependencies between sequential actions during tool use, and if so (2) to determine whether the distance at which nonadjacent dependencies occurred in action sequences, and the dynamics of mutual information decay between increasingly separated elements, supports existing claims that chimpanzees hierarchically organize their action sequences during tool use. To answer both of these questions, we coded 8,261 actions of eight wild chimpanzees at Bossou, Guinea, as they spontaneously cracked 303 individual nuts using stone tools.

For six out of eight individuals, including two-thirds of adults (we take adulthood to be a proxy for experience and proficiency in tool use), nonadjacent dependencies were identifiable at sequential distances which significantly exceeded those produced in sequences simulated from Markov models. Thus, our data suggest that between the

sequential actions produced by multiple chimpanzees during stone-tool use, nonadjacent dependencies occur at distances which exceed those found in comparable non-hierarchical sequences. For half of adults, the extended distances at which we detected dependencies could not be solely explained by repeating actions; however for the other half of adults, as well as for both the juvenile and infant individuals in our analysis, repeating actions contributed substantially to the emergence of nonadjacent dependencies. This finding does not preclude these individuals from generating some dependencies through supraregular means of action sequence generation, such as hierarchical structuring (see further discussion on MI decay below); however, it does suggest that repeated actions contributed to a greater proportion of nonadjacent dependencies which spanned larger sequential distances. These results therefore suggest that there is detectable interindividual variation in the means through which nonadjacent dependencies are generated within the action sequences of different chimpanzees, particularly between distantly separated actions.

Interindividual differences in the distances at which dependencies occurred in action sequences is consistent with previously described differences in nut cracking techniques between chimpanzees at Bossou (*Berdugo et al., 2023*). Additionally, our analysis reveals that younger individuals may be more likely to produce dependencies through repeating actions, such as in the case of Flanle who showed a drastic decrease in the maximum distance dependencies detected after sequence compression. As the youngest individual in our analysis, Flanle was highly inexperienced in nut cracking, and we only observed Flanle successfully cracking open a single nut during the entire period of observation (for contrast, the other seven individuals for our analysis cracked a total of 302 nuts; mean = 43 nuts each, SD = 26.9). Flanle demonstrated the highest number of action types (59), indicating a greater diversity of different action types compared to the action sequences of adult chimpanzees during nut cracking (for whom action sequences were highly stereotyped in their use of a small number of different actions). This diversity of actions, as well as their repetition and flexible assembly, may have contributed to the anomalous distance at which dependencies were identified in the action sequences for Flanle. Further research is required to understand how these factors influence the emergence of sequential dependencies in animal behaviours.

Once repeating actions were controlled for, half of adult individuals in our analysis produced dependencies at distances which exceeded comparable Markov models. This result suggests that nonadjacent dependencies are produced by a substantial proportion of chimpanzees who are proficient in tool use, at distances which exceed expectations if sequences were generated through non-hierarchical means. Among the sequential behaviours of wild chimpanzees, tool use is the first domain in which such nonadjacent dependencies have been detected. This data provides new context to results from experimental paradigms which have previously revealed that captive chimpanzees are able to identify nonadjacent dependencies in both visual and auditory sequences (*Sonnweber, Ravignani & Fitch, 2015*; *Watson et al., 2020*), by offering an ecologically-relevant domain of behaviour in which such cognition can be used. It is possible that, within the domain of manual actions, tool use is a unique behaviour of wild chimpanzees which leads to the production of nonadjacent sequential dependencies. In another study performed on the

sequential play behaviours performed by chimpanzees at Bossou, the accuracy with which an action in a play sequence could be predicted was aided by considering the two most recent actions, but actions further back in the sequence offered no additional benefit (*Mielke & Carvalho, 2022*). This suggests that play behaviours do not produce long-range dependencies, such as those we have identified during tool use. However, to validate whether tool use is unique in its production of nonadjacent dependencies, it will be necessary to analyse a wider range of sequential manual behaviours of wild chimpanzees, and also to ensure that comparisons are made with data collected and analysed using directly comparable methodologies. These analyses should include sequential manual behaviours which vary in their technical complexity, and should include both complex, multi-stage manual behaviours (*e.g.*, nest building and complex foraging tasks which do not require tools), as well as simpler feeding tasks which require fewer processing steps (*Gontier, 2024*).

The sequential manual actions of non-human apes—including those used in tool use—have previously been described using qualitative frameworks which either explicitly or implicitly encompass hierarchical organization (*Byrne & Russon, 1998*; *Carvalho et al., 2008*; *Byrne, Sanz & Morgan, 2013*; *Estienne, Stephens & Boesch, 2017*; *Boesch et al., 2020*; *Gontier, 2024*); however, it has thus far been empirically challenging to exclude the possibility that these action sequences are produced by simpler structuring mechanisms, such as serial chaining (*Vereijken & Whiting, 1998*), particularly as human observers carry cognitive biases for identifying hierarchical structures in non-hierarchical sequences (*Fitch, 2014*; *Ferrigno et al., 2020*). Through characterizing MI decay across observed action sequences, we identified a period of power-law decay for the majority of chimpanzees. Whilst our data is observational, we can confirm that this result matches the MI decay dynamics predicted by hierarchical organization of action sequences. Of the chimpanzees whose MI decay profiles included periods of power-law decay, the composite model was more frequently selected as the optimal model. We therefore found the greatest support for composite structuring systems, where actions are organized into short subroutines through association rules between neighbouring elements, which were then hierarchically organized to produce longer sequences of tool-action. Through analysing the curvature of composite models, we identified that these subroutines of actions were likely between 2–8 elements long. Our results are therefore consistent with previous descriptions of how great apes organize sequences of actions during food processing behaviours, including those which involve the use of tools (*Byrne & Russon, 1998*; *Byrne, Sanz & Morgan, 2013*; *Gontier, 2024*). Apes are theorized to decompose goals into subgoals which are addressed through short, stereotyped subroutines of actions (*Byrne & Russon, 1998*; *Byrne, Sanz & Morgan, 2013*), thus, these models of behavioural organization are identical to the composite structuring model for which our analysis provided the greatest support. Additionally, our analysis is able to objectively offer support to the hypothesis that apes organize sequences through the hierarchical organization of subroutines, without the presence of human bias for identification of hierarchical phenomena through subjective descriptions.

Despite the prevalence of the composite model in characterizing MI decay, our analyses did not provide conclusive results for all chimpanzees, as we could not resolve for a single best model for two individuals once we had accounted for repeating actions. For one individual, Foaf, the best model was identified to be the exponential model, which is associated with sequences which are produced using serial chaining. Such interindividual differences may be further evidence of differences in strategies for nut cracking behaviours between individuals, with a minority of individuals relying on alternative, finite-state mechanisms of sequence generation. However, it is also important to recognize that for Foaf—the only individual for whom the exponential model was solely preferred—a much smaller corpus of actions was collected compared to those of other individuals (see Table 1). We are not able to say for certain whether sample size would have influenced the model preference for Foaf. However, given that sample size can influence entropy estimations (*Grassberger, 2003*), we can only conclude that this is a possibility for our analysis. We recommend further research be conducted to investigate possible interindividual variability in the systems of sequential organization employed by chimpanzees.

Whilst our analysis provides the strongest evidence for composite mechanisms of action organization in chimpanzee tool use, it is important to emphasize that our study is observational. It is therefore outside of the scope of this study to determine what parsing rules chimpanzees may use to facilitate a structuring system which relies on regular and supraregular dynamics at different temporal scales (a limitation recognized in similar analyses on birdsong; *Sainburg et al., 2019*). For example, whilst we provide a prediction for the average length of subroutines (2–8 actions), we cannot identify the precise actions that subroutines contain, nor the amount of variation in subroutine length within sequences. Additionally, our study is unable to provide insight into the exact representational mechanisms and parsing procedures that chimpanzees use to produce successful sequences of tool action (*Gruber et al., 2015*). This includes the extent to which chimpanzees have conceptions of 'parent-offspring' relationships between higher-order goals and their constituent elements, or whether hierarchical patterns could emerge from comparatively simpler systems, such as those similar to stack-memory in automata that can generalize phrase-structure sequences (*Ong, 2007*). In the latter case, chimpanzees may construct hierarchical patterns by pausing the performance of action sequences directed at one goal, to embed in shorter subroutines of actions for corrective behaviours, before returning to the completion of the original sequence (for example, pausing the striking of a nut with a hammer to reposition a nut on the anvil, or to rotate stones into a more stable configuration). These limitations also extend to the degree to which working memory is employed to account for distant dependencies. For example, it is possible that during nut cracking, physical traces of previous actions are maintained in the environment, rather than in the mind, reducing demands on memory (*e.g.*, a nut positioned on top of an anvil would suggest it has already been placed there in a previous action; *Byrne & Russon, 1998*). Furthermore, whilst the hierarchical organization of subroutines has long been theorized to be the most likely form of sequential action organization in chimpanzees, it is possible

that a small number of alternative supraregular systems of sequence generation may generate nonadjacent dependencies (see *Wilson et al., 2020*). We did not include these models of sequence generation in our analysis for two reasons. Firstly, we do not currently have clear predictions of how these mechanisms of sequence generation would lead to mutual information decay over increasing sequence length. Secondly, these systems of sequence generation are not supported by theoretical literature to the same degree as hierarchical models of action organization for tool use. We therefore restricted our analysis to comparing models of sequence generation that are informed by substantial theoretical support, and conclude that the patterns of dependencies produced by chimpanzees generally follow those which are predicted by the hierarchical arrangement of subroutines. We strongly recommend alternative supraregular grammar systems are tested where and when suitable.

In contrast with other studies which employ analyses of MI decay (*Sainburg et al., 2019*; *Sainburg, Mai & Gentner, 2020*, *2022*; *Youngblood, 2024*), we used sequence data from Markov models to identify the rate at which power-law relationships would be identified from linear sequences. This offered a false positive rate for misidentifying signatures of hierarchical organization. Power-law relationships were identified from linear sequences at a non-negligible rate, indicating that using MI decay analyses to identify signatures of hierarchy can produce false-positive results. However, this false positive rate was substantially lower than the rate at which power-law relationships were identified in observed action sequences. We therefore can conclude that the power-law dynamics identified in chimpanzee action sequences did not represent false positives. Instead, power-law dynamics in chimpanzee action sequences more likely reflect that actions are not combined by systems akin to Markovian sequence generation. Moreover, we strongly advise that future research employing MI decay analyses employ Markov models to estimate false positive rates of power-law detection. These false positive rates can then be compared with the rates of power-law detection from real-world sequential data to ensure that power-law model preferences are not statistical artefacts.

MI estimation has previously been used to detect signatures of hierarchical organization in the intricate grooming behaviours of Drosophila flies (*Drosophila melanogaster*) over an hour, and the swimming behaviours of Zebrafish (*Danio rerio*) during phototaxis paradigms (*Sainburg, Mai & Gentner, 2020*), and for both species, the composite model was the best choice for explaining information decay over the action sequences of individuals. Given our analysis found the greatest support for the composite model, we believe that our results likely reflect a possible homologous structuring system between the tool use behaviours of chimpanzees during nut cracking, and in the organization of goal directed actions by other species, when performing sequential actions for behaviours which do not involve tools. Additionally, the hierarchical complexity of early hominin tool-related action is believed to have increased throughout the course of hominin evolution, with the production of more ancient Oldowan tools requiring less complex hierarchical organization than the production of more recent Acheulian stone tools (*Stout et al., 2021*). In light of our results, it is possible that the hierarchical organization of tool-use behaviours may have been an adaptation which was present in the evolutionary

ancestor of humans and apes—stemming from more basal hierarchical systems of action—which was then elaborated across the course of human evolution into a state of higher complexity. Understanding the dynamics of this evolutionary transition would be greatly aided through further comparative research into the sequential organization of actions used by chimpanzees and humans during a greater variety of tool-use tasks, as well as everyday behaviours which require careful sequencing of manual actions (*Gontier, 2024*). This includes establishing whether nonadjacent dependencies emerge in other manual behaviours of great apes, and if so, whether these nonadjacent dependencies are the product of hierarchical organization.

As previously mentioned, outside of tool use, the presence of hierarchical thinking in primates may also manifest within alternative behavioural contexts and social cognition, including in species other than chimpanzees (*Bergman et al., 2003*; *Mielke & Carvalho, 2022*; *Lameira et al., 2024*). In all of these instances, comparative research across domains of behaviour would be of interest, both within and between species. This research will be useful for determining whether hierarchical behaviours share common cognitive resources, or if these behaviours involve fundamentally different conceptions of hierarchies (see similar arguments made in human behaviours; *Coopmans, Kaushik & Martin, 2023*). For example in apes, comparisons of chimpanzee play sequences and tool use may be informative for understanding when nonadjacent decencies can be expected in sequential manual actions, and how this expectation can be linked to fast-changing goals (such as in play behaviours) or goals which are sustained over extended periods of time (such as in tool use). Additionally orangutans offer an excellent opportunity to investigate whether the hierarchical complexity of vocal sequences, and the sequences of manual actions performed during tool use, covary between individuals. Orangutans would be particularly suitable for such an analysis, given their habitual use of hierarchically organized long calls (*Lameira et al., 2024*), and that they spontaneously manufacture and use tools in the wild (*Van Schaik, Fox & Sitompul, 1996*).

Further research comparing the structure of sequential behaviours within ape species will clarify the extent to which these putatively hierarchical phenomena are supported by domain general cognition. This question is currently central to a number of hypotheses surrounding the evolution of cognition for hierarchical syntax and nonadjacent dependencies in human behaviours, such as language and tool use. For example, the statistical scaffolding hypothesis (*Sainburg, Mai & Gentner, 2020*) posits that long-range statistical dependencies in non-linguistic behaviours—and a generalised sensitivity to such behaviours—has led to a scaffold onto which language could evolve, where complex syntax and semantics can be understood as later additions that exploit such structures and sensitivities. Similarly, the Cognitive Coupling hypothesis suggests that the pre-existing cognitive architecture responsible for hierarchical tool behaviours was coupled with cognitive machinery responsible for communication, where such coupling occurred in multiple steps across hominin evolutionary history, driven by the need to socially learn and transmit increasingly more complex tool behaviours (*Kolodny & Edelman, 2018*).

We believe that the findings of our study are consistent with, but not confirmatory of, both the statistical scaffolding hypothesis, and the cognitive coupling hypothesis. We have

presented data herein that supports the presence of nonadjacent dependencies in the sequential action of chimpanzee tool use, as well as decay dynamics of MI that are readily generalised by models associated with the hierarchical organization of action. Evidence for these structural features of sequential organization in chimpanzee tool use imply homologous origins with the sequential organization of actions in humans, suggesting that these structural features may also have been present in their last common ancestor. The emergence of highly hierarchical organization within communicative behaviours, and the resultant production of frequent nonadjacent dependencies in communication, may then have evolved as human-specific traits, following the evolutionary divergence of the human and *Pan* lineages. Further empirical investigation into the sequential organization of calls and gestures in chimpanzees and bonobos would offer further clarity on this matter.

## CONCLUSION

In sum, we have identified that chimpanzees produce nonadjacent dependencies within their sequential tool-use behaviours, and exhibit interesting interindividual variability in the distances at which dependencies occur and in the means through which they are generated. These dependencies include those which extend past the maximum distance dependencies that were detected in sequences generated by Markov models, which for half of adult chimpanzees, cannot be explained by repeated actions. Our results of MI decay also support previous descriptions that chimpanzees organize their actions during tool-use through the hierarchical arrangement of subroutines of actions, and through analysing the curvature of composite decay models, we estimate that these subroutines are between 2–8 actions in length. However, we also note that this result was not universal to all individuals, with MI decay following alternative patterns for several individuals. We discuss this result in light of possible interindividual differences in systems of sequence organization by chimpanzees during tool use. Further research will determine whether primates' other sequential behaviours exhibit nonadjacent dependencies similar to those we have found in tool use, and will clarify that these dependencies are generated through hierarchical means. This research will reveal the true extent to which nonadjacent dependencies, and the hierarchical systems of sequence organization that we provide evidence for, are the product of domain general cognition in nonhuman primates. In extension, this research will be informative for understanding whether the cognition supporting the generation of processing of nonadjacent dependencies in human behaviour, including through hierarchical structuring, were exapted across behavioural domains during hominin evolution.

## ACKNOWLEDGEMENTS

We thank Susana Carvalho for their help in curating the Bossou video archive, and Tim Sainburg for statistical advice. We thank Anna Ahlberg and Melissa Birtles for offering their time to test the repeatability of our ethogram. The authors would like to acknowledge the use of the University of Oxford Advanced Research Computing (ARC) facility in carrying out this work (http://dx.doi.org/10.5281/zenodo.22558). We are also grateful to all the KUPRI researchers who have helped collect data at Bossou in the past. Special

thanks are due to Direction General de la Recherche Scientifique et l'innovation Technologique (DGERSIT) and the Institut de Recherche Environnementale de Bossou (IREB), République de Guinée, for facilitating field work at Bossou; as well as research assistants Boniface Zogbila, Gouanou Zogbila, Henry Gbelegbe, Marcel Doré, and Gilles Doré for their help in the field.

### Funding

Elliot Howard-Spink was supported by National Environmental Research Council (NERC) and United Kingdom Research and Innovation (UKRI; grant ref. NE/L002612/1). Thibaud Gruber was supported by the Swiss National Science Foundation (SNSF; grant ref. PCEFP1_186832). Tetsuro Matsuzawa was supported by grants from Ministry of Education, Culture, Sports, Science and Technology, Japan (MEXT; #12002009, #16002001, #20002001, #24000001, #16H06283) and Japan Society for the Promotion of Science (JSPS; Core-to-core CCSN and U04-PWS). Misato Hayashi was supported by grants MEXT/JSPS 15K00204, 19K21824, and JP17H06381 (Evolinguistics). Daniel Schofield was supported by grants from The Clarendon Fund, Boise Trust Fund, and Wolfson College, University of Oxford. The funders had no role in study design, data collection and analysis, decision to publish, or preparation of the manuscript.

### Grant Disclosures

The following grant information was disclosed by the authors:
National Environmental Research Council (NERC).
United Kingdom Research and Innovation (UKRI): NE/L002612/1.
Swiss National Science Foundation (SNSF): PCEFP1_186832.
Ministry of Education, Culture, Sports, Science and Technology, Japan (MEXT): #12002009, #16002001, #20002001, #24000001, #16H06283.
Japan Society for the Promotion of Science (JSPS).
MEXT/JSPS: 15K00204, 19K21824, JP17H06381.
The Clarendon Fund.
Boise Trust Fund.
Wolfson College.
University of Oxford.

### Competing Interests

The authors declare that they have no competing interests. Tetsuro Matsuzawa is an Academic Editor for PeerJ.

### Author Contributions

- Elliot Howard-Spink conceived and designed the experiments, performed the experiments, analyzed the data, prepared figures and/or tables, authored or reviewed drafts of the article, and approved the final draft.

- Misato Hayashi conceived and designed the experiments, performed the experiments, authored or reviewed drafts of the article, organized access to the Bossou archive, and approved the final draft.
- Tetsuro Matsuzawa conceived and designed the experiments, performed the experiments, authored or reviewed drafts of the article, organized access to the Bossou archive, and approved the final draft.
- Daniel Schofield conceived and designed the experiments, authored or reviewed drafts of the article, processed, cleaned and curated the Bossou archive, and approved the final draft.
- Thibaud Gruber conceived and designed the experiments, authored or reviewed drafts of the article, supervisor, and approved the final draft.
- Dora Biro conceived and designed the experiments, performed the experiments, authored or reviewed drafts of the article, supervisor, and approved the final draft.

## Animal Ethics

The following information was supplied relating to ethical approvals (*i.e.*, approving body and any reference numbers):

The collection of video data of wild chimpanzees at Bossou was non-invasive, and approved by the ethics committee at the Primate Research Institute, Kyoto University.

## Data Availability

All data and code are available at Mendeley Data: Howard-Spink, Elliot; Hayashi, Misato; Matsuzawa, Tetsuro; Schofield, Daniel; Gruber, Thibaud; Biro, Dora (2024), "Data: Nonadjacent Dependencies and Sequential Structure of Chimpanzee Action During a Natural Tool-Use Task", Mendeley Data, V3, DOI: 10.17632/xdtnxzdchj.3.

## Supplemental Information

Supplemental information for this article can be found online at http://dx.doi.org/10.7717/peerj.18484#supplemental-information.

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
