# Peer review of "Nonadjacent dependencies and sequential structure of chimpanzee action during a natural tool-use task"

_PeerJ, doi:10.7717/peerj.18484_

## Round 0.1 · original submission · Major Revisions

Dear Dr Howard-Spink and colleagues, Many thanks for submitting your manuscript to PeerJ. The manuscript has now been reviewed by two reviewers. Following their comments and my own reading of the paper, I am recommending major revisions. While both reviewers recognize and highlight the merit of the paper, they made some comments, especially regarding terminology as well as data analysis and interpretation that you should address before the manuscript can be considered for publication.

Finally, when preparing the manuscript for resubmission, make sure you follow the PeerJ author guidelines more closely, by, for instance, inserting the methods right after the introduction and by sharing the data and R-codes. You can find more details on the required submission format here https://peerj.com/about/author-instructions/

Many thanks again

Reviewer 1 ·

Basic reporting

Data and code not provided

I’m a bit unclear on what are the 82 classes of action, because I don’t think there are 82 behaviours in the ethogram in the supplementary materials

I feel you should report the confidence intervals for the observed sequences also in figure 2. If the confidence intervals overlap I think this will effect the findings a lot.

What is the interpretation of differences between the full and condensed models?

I find the interpretation of the transition point models a bit difficult. Is it after 2-8 elements then they start hierarchically structuring or is it that there are strings of 2-8 elements that are put together in combinations to produce hierarchical structures? And how do we know?

Line 440 – you should either remove the second use of the word ‘of’ at the beginning of this sentence or add the word ‘the’ after.

Line 447 – should be 'individuals'

Line 481 – should be ‘tool use is a unique behaviour’

Line 686 – I’m wondering why just that year? Bossou has been researched for decades and one of the major limiations of the study is the small amount of footage, so why not make more use of what is presumably available in the archive?

Lines 780 – 785 – Not clear to me how this deals with the problem of MI decaying to zero and also not clear why decaying to zero before the maximal comparative distance is a problem. I’m sure it’s totally valid but it’s just not explained in enough details for me (or a reader) to understand.

Line 812 – I think it would be good to specify the other terms in the equation as well as x.

Lines 876 – 879 – So I am still a bit confused on whether you did the markov chains with including repetitions and non-repetitions or not. Because in the results it’s not clear but there is only one community markov model used to compare, but I feel like it’s important to have a community model with repetitions to compare to observed model with repetitions and you should then use the same markov data but condensing the repetitions to compare to your condense sequences.

Some of the supplements are not discussed in the text (supplementary figs 2 and 3 as well as supplementary tables 4 and 5 for example) so I guess you maybe don’t need to include them

Experimental design

No major remarks on the design, I think in principle everything is adequate to meet the aims of the study.

Validity of the findings

I think the results are not super convincing. Firstly, the finding that two thirds showed non-adjacent dependencies is not necessarily true because the 95% CI for the observed sequences are not given so we don’t know if they overlap with the markov model estimates. Assuming they were true, with and without the repeated actions many individuals change preferred model type and actually when the action sequences are condenced only 3 individuals show an estimate outside of the 95% CI for the markov model and really only marginally so. If you also plotted the CI’s for those cases I’m certain this would overlap in 2 of the cases (fanle and tua), but potentially all 3. This is made more worrying by the fact that in line 407 it is claimed that if this was being generated by non-hierarchical markov models we would expect composite power law models to be the best in 3/8 cases, which is exactly what you find when the sequences are condensed. The could be saved by the fact that the Markov models include repetitions, but it’s not clear whether this is the case or not.

Lines 746-747 – I’m wondering if concatenating all the sequences per individuals is leading you to over-estimate dependencies? For example, if you usually finish on a certain behavior (such as eat nut) and usually start on a certain behavior (place nut on anvil perhaps), then this is probably going to create a dependency that is totally artificial, because those sequences could be totally independent from each other. How exactly does the concatenation work? Is it all the sequences in one bout or is it all the sequences ever observed during the observation period? Because in the latter case I think it might be a problem.

Additional comments

I feel a bit that the manuscript goes between talking about non-adjacent dependencies and hierarchical structuring, which as the authors say, are related, but are not the same thing because while it is true that hierarchical structuring involves non-adjacent dependencies, not all non-adjacent dependencies are hierarachically structured. I do understand that hierarchical structuring is indeed implied by the observed decay overtime, but to show this would require more work, maybe even a different approach. In general I think the authors should stick to talking about non-adjacent dependencies about emphasise that this is really only indirect evidence of hierarchical structuring in the discussion.

Lines 79-83 – I feel that the reasoning is the wrong way around here. Hierarchical structuring presupposes non-adjacent dependencies. But non-adjacent dependencies do not presuppose hierarchical structuring. The sequence A -> X -> B where B is dependent on A doesn’t imply a hierarchy, no matter how many X’s separate A and B, because nothing is ‘beneath’ anything else. X isn’t a sub-section of the overall sequence, it’s just intervening between A and B. There are many papers that explain this, but here is one nice one that you already cite and I think could be helpful: Wilson, B., Spierings, M., Ravignani, A., Mueller, J. L., Mintz, T. H., Wijnen, F., ... & Rey, A. (2020). Non‐adjacent dependency learning in humans and other animals. Topics in cognitive science, 12(3), 843-858.

Reviewer 2 ·

Basic reporting

No comment

Experimental design

No comment

Validity of the findings

No comment

Additional comments

I have reviewed the paper by Howard-Spink et al.
This paper investigates the sequential structuring of tool-use behaviour (nut cracking) in wild chimpanzees. Using cutting-edge MI-based approaches and critical simulated Markov models as controls, the authors convincingly demonstrate signatures of hierarchical organization in these nut-cracking action sequences.
This paper represents an extremely important advance and contribution to the field not only methodologically but also empirically, providing some of the most robust evidence for hierarchical structuring in great ape behaviour to date. I strongly recommend publication in PeerJ and have a few minor-ish points that it would be great if the authors could consider in a revised version.
Major points:
1) Syntax is a loaded term particularly because there are quite some differences in the field with regards to what this term means precisely. If the authors want to use this term in the paper then I think they need to provide an explicit definition as to what they mean by syntax and potentially that there are other interpretations/definitions. Furthermore, I would perhaps refrain from focusing on this too heavily in the title and rather talk more about “sequential structure” (but ultimately, I leave this decision up to the authors).
2) L155: I think there is some research in Turkish suggesting NADs exist at the phonemic level, so perhaps qualify this statement with “though see x”.
3) L263: Why did you not focus the analysis solely on adults? Is it not the case that adding individuals from different age classes generates additional noise that might mask or influence your findings?
4) L486: The differences you raise with the sequencing of play behaviour are interesting. Could any observed differences also be driven by the use of different metholodologies to identify long-distance dependencies? Maybe this could be mentioned briefly?
5) L620-622: There already exists some tentative evidence for internal structuring of chimpanzee vocal sequences (e.g. nested structure (bigrams+unigrams), see Giraud-Buttoz et al. 2022). Depending on one’s definition of hierarchy, it could be argued that very basic dependencies such as these might constitute evidence for hierarchical structuring in non-human primate communication systems. Perhaps the authors could mention or address this?
6) L626: I find the potential inter-individual differences in hierarchical structuring very interesting. Is there any evidence in humans for example that there is consistent variation in the hierarchical structuring of complex behaviour (action or language) between individuals? If so, this might be useful and relevant to bring up here.


Minor points:
L67: Could delete the sentence “such as the paragraphs you are reading now”.
L110: “Highly hierarchical” reads a bit odd. Maybe “complex hierarchical”?
L127: serial chains? Or reflexive serial chains?
L132: replace “highly” with “significantly”
L134: “replicated” or would “captured” be a better phrase?
L134-137: this link to Markov chains comes a bit out of the blue. Maybe the transition could be improved by referring more generally to simple stochastic modelling approaches such as Markov chains?
L186: Issue with the grammar? Should it be “during” or “in” but not both?
L394: It seems that a word is missing here? Please check.
L477: Change “results” to “data” to avoid repetition.
L481: “Is a unique behaviour”?
L559: Not clear why you start with “However” here since this is not really contrasting your point but you are advocating other researchers to use the same approach as you. Rephrase?
L632: Should this be “including for example”

---

## Round 0.2 · Major Revisions

Dear Dr Howard-Spink and colleagues,

Many thanks for revising your manuscript according to reviewers' comments. I have resent your revision to one of the original reviewers. While they are happy with how you addressed their original comments, they wanted to see some additional changes, especially in regards to how some of the results are interpreted. I would recommend addressing these additional comments before considering accepting the manuscript for publication.

Reviewer 1 ·

Basic reporting

All previous concerns in this aspect have been addressed but there were a couple of points that I think could be better reported in the introduction.

At line 100 the authors allude to similarities between language and action, but this is in fact not a settled issue at all. Some do indeed argue as you say that there strong similarities, others outright deny that the structure of language and action have anything to do with each other. I feel like this debate should be highlighted more clearly.

Line 103 - heierchical organization presumes nonadjacent dependencies and since there are non-adjacent dependenceis without hierarchies but not hierarchies without nonadjacent dependncies I don’t think it makes sense to say that nonadjacent dependencies emerge or result from hierarchies.

Line 114 - What do we really know about the underlying neural architecture, how it compares to the architecture underpinning simple sequences with no long-distance dependencies, and what that means in terms of costs?

Line 146 - I feel like this is really using hierarchy in a very loose sense. What this paper is talking about in the end is really just hierarchical organization of temporal sequencies, the type of hierarchy this reference refers to is something totally different.

Lines 140-165 - This whole discussion is really just focusing on the wrong thing I feel, the topic is non-adjacent dependencies.

Experimental design

One question that came to mind this time around is why there is not an analysis of the transition points of markov-stimulated sequences for the composite models? I'm still struggling to understand somewhat these composite models, but from what I have understood, you find that at short distances exponential models are the best fit and then at longer distances power-law models are a better fit, but since markovian dynamics only work at short-distances, wouldn't this always be the case? I suppose I am wondering whether at short distances power-law decay models and exponential decay models make different predictions? I thought looking at the transition point for composite models that fit the markov-simulated sequences (of which there are actually quite a lot - 1/4 markov-simulated sequences are best described by a composite model if I have understood correctly) might help to better understand what these composite models of the chimp data are really showing.

I'm also still not convinced about the interpretation of the composite models. If they show that at short distances exponential decay is the best fit, then as you get further in the sequence power-law decay is the best fit, how does this show that those short sequences are being hierarchically organized into more complex structures? doesn't that assume that those behaviours later in the sequence are the same behaviours from earlier in the sequence repeated and organized into some more complex structure? Also wouldn't the unit of analysis change? If behaviours are organized hierarchically, those behaviours should form a single unit and then it is that unit that is being organized into more complex structures and I really don't see how this shows that. In fact this approach is pretty much opaque about what the actual behaviours are. In the end what we're looking at here is the mutual information between states in the sequence, so it just means that there are long distance dependencies, but I don't see why it means the first 2-8 actions in the sequence are being organized into a hierarchy?

Validity of the findings

I still feel the focus is just way to heavy on the wrong thing, especially in the discussion. What this is really about is non-adjacent dependencies, and not hierarchical structuring. Hierarchical structuring is one of the possible mechanisms through which such dependencies can be produced, as shown by Lin & Tegmarks work which you cite. However, as the authors wrote in their response and allude to in their discussion, other mechanisms such as stack (or push-down stack) memory can generate such dependencies from a computational perspective, but these mechanisms do not involve hierarchical structuring. In the case of a stack memory interpretation, a stack is a linear structure, it does not have parent-child relationships as we find in hierarchical structures. As such, I feel like these different possible interpretations should be equally discussed. Currently, these alternatives are only really briefly pointed out in line 692. Later in the discussion (line 809) the authors argue that they focus on hierarchical structuring because we don't have models with clear predictions for other mechanisms, but I think this is mischaractarising the problem. The problem is not that you have an approach that definitively shows hierarchical structuring but you don’t have a model to test for other mechanisms that can generate non-adjacent dependencies, the problem is that you have a model that is telling you there are non-adjacent dependencies, but through what mechanism (hierarchical structuring, push-down stack) we just don’t know. So I feel like the alternatives should be equally discussed at this point, but the discussion is very heavy towards hierarchical structuring.

Additional comments

This is a really nice paper and it's very interesting, but I feel it's focus is just misdirected. The topic is non-adjacent dependencies, and what the analysis shows (very nicely) is non-adjacent dependencies. There are several possible interpretations of those dependencies, hierarchical structuring, push-down stack, and I'm sure many others also. I am not convinced that the paper shows evidence for any of these interpretations in particular, they are all possible, and the next step would be to try and discriminate between them, but it's not what is done here so I feel the interpretation and discussion needs to be much more even handed in this sense.

---

## Round 0.3 · Minor Revisions

Dear Dr Howard-Spink and colleagues. Many thanks for revising your manuscript. I have sent your revision to one of the original reviewers who is in general happy with how you addressed their comments. However, they provide some additional minor comments, that I would like you to address before formally accepting the manuscript.

Reviewer 1 ·

Basic reporting

All good!

Experimental design

No problems with the design.

Validity of the findings

I accept the arguments that even at short distances exponential and power-law models make different predictions (although I still imagine the size of those differences at short distance is presumably much smaller then at longer distances and with the small sample in hand it’s probably not detectable at such distances – a point that’s maybe worth highlighting somewhere) and that while we don’t really know the relationship between Markovian systems and composite models, it is at least true in this sample that composite models are more commonly a better fit than in the corresponding pure-markovian models.

Regarding the interpretation of the composite models – the more detailed explanation is certainly helpful and gives me a much better understanding of what’s going on here, but I still feel that their explanation is not consistent with their interpretation. In the response, the authors say ‘this is not a question of the first 2-8 actions being then turned into a hierarchy’, yet in the abstract say ‘Our analysis offered the greatest support for a system of organization which involved the production of short subroutines of actions (2-8 actions), which are hierarchically arranges into sequences’. This seems contradictory and either the inference that is drawn from the analysis should change, or, a better argument needs to be given for why this inference is valid.

I will give an example. In the attached PDF are two examples of potentially ‘composite systems’ as I understand them (as systems characterized by a combination of Markovian adjacent dependencies and [potentially] hierarchically organized non-adjacent dependencies). In the first example (A), you have a sequence of elements on the left wherein the first state is a daughter of a parent node, but from there proceeds according to Markovian adjacent dependencies (as seen in Lin and Tegmark’s example of ‘shallow dynamics’), while subsequent states in the sequence are governed by higher-order states (i.e., every state is a ‘daughter’ of some higher-order node). In the second example (B), we see three branches organized under a higher order state, two of which are grouped under the same sub-node, but within each we see a sequence that proceeds according to markovian adjacent dependencies as in the first part of the sequence in the first example. I think the latter example is the inference the authors draw from their models (i.e., markovian short sequences embedded within a hierarchy), but aren’t both of these examples compatible with the observed patterns of MI decay that a composite model generates? If so I feel like the best that can be said is that both types of dynamics seem to be present in the system, but how exactly they are arranged we don’t know (and maybe can’t know exactly when using stochastic methods?).

Additional comments

It's a great manuscript and I appreciate the changes the authors have made. The discussion is still a bit heavy on the hierarchical structure interpretation but at this point I don't think our conversation on this issue is really going anywhere so we can just agree to disagree, that's fine. The only remaining issue I really have is a bit of inconsistency/ambiguity surrounding the interpretation - to me I think there are multiple types of composite systems consistent with the observed patterns and we can't really tell which one it is with this approach. Probably this can just be addressed by simply saying the composite system 'could' involve 2-8 sequence long routines being reorganized, but it's also possible that individuals switch to an entirely different (non-markovian) dynamic after 2-8 actions.

Annotated reviews are not available for download in order to protect the identity of reviewers who chose to remain anonymous.

---

## Round 0.4 · accepted · Accept

Dear Dr Howard-Spink and colleagues,

Many thanks for revising the manuscript and addressing the final minor comments by the reviewer. After reading your revision, I am happy to accept the manuscript in its current form. Congratulations!